# Human skeletal muscle organoids model fetal myogenesis and sustain uncommitted PAX7 myogenic progenitors

Lampros Mavrommatis[1,2,3]*, Hyun-Woo Jeong[4]*, Urs Kindler[1], Gemma Gomez-Giro[5], Marie-Cecile Kienitz[6], Martin Stehling[7], Olympia E Psathaki[2,8], Dagmar Zeuschner[9], M Gabriele Bixel[10], Dong Han[2], Gabriela Morosan-Puopolo[1], Daniela Gerovska[11], Ji Hun Yang[12,13], Jeong Beom Kim[14], Marcos J Arauzo-Bravo[11], Jens C Schwamborn[5], Stephan A Hahn[15], Ralf H Adams[10,16], Hans R Schöler[2], Matthias Vorgerd[3], Beate Brand-Saberi[1], Holm Zaehres[1,2]*

[1]Ruhr University Bochum, Medical Faculty, Institute of Anatomy, Department of Anatomy and Molecular Embryology, Bochum, Germany; [2]Max Planck Institute for Molecular Biomedicine, Department of Cell and Developmental Biology, Münster, Germany; [3]Department of Neurology with Heimer Institute for Muscle Research, University Hospital Bergmannsheil, Bochum, Germany; [4]Max Planck Institute for Molecular Biomedicine, Sequencing Core Facility, Münster, Germany; [5]Luxembourg Centre for Systems Biomedicine, LCSB, Developmental and Cellular Biology, University of Luxembourg, Belvaux, Luxembourg; [6]Ruhr University Bochum, Medical Faculty, Department of Cellular Physiology, Bochum, Germany; [7]Max Planck Institute for Molecular Biomedicine, Flow Cytometry Unit, Münster, Germany; [8]Center for Cellular Nanoanalytics Osnabrück, CellNanOs, University of Osnabrück, Osnabrück, Germany; [9]Max Planck Institute for Molecular Biomedicine, Electron Microscopy Unit, Münster, Germany; [10]Max Planck Institute for Molecular Biomedicine, Department of Tissue Morphogenesis, Münster, Germany; [11]Computational Biology and Systems Biomedicine, Biodonostia Health Research Institute, San Sebastián, Spain; [12]School of Mechanical Engineering, Korea University, Seoul, Republic of Korea; [13]R&D Research Center, Next & Bio Inc, Seoul, Republic of Korea; [14]School of Life Sciences, Ulsan National Institute of Science and Technology (UNIST), Ulsan, Republic of Korea; [15]Ruhr University Bochum, Medical Faculty, Department of Molecular GI Oncology, Bochum, Germany; [16]Westphalian Wilhelms University Münster, Medical Faculty, Münster, Germany

*For correspondence:
lampros.mavrommatis@rub.de (LM);
hyun-woo.jeong@mpi-muenster.mpg.de (H-WJ);
holm.zaehres@rub.de (HZ)

**Abstract** In vitro culture systems that structurally model human myogenesis and promote PAX7[+] myogenic progenitor maturation have not been established. Here we report that human skeletal muscle organoids can be differentiated from induced pluripotent stem cell lines to contain paraxial mesoderm and neuromesodermal progenitors and develop into organized structures reassembling neural plate border and dermomyotome. Culture conditions instigate neural lineage arrest and promote fetal hypaxial myogenesis toward limb axial anatomical identity, with generation of sustainable uncommitted PAX7 myogenic progenitors and fibroadipogenic (PDGFRa+) progenitor populations equivalent to those from the second trimester of human gestation. Single-cell comparison to human fetal and adult myogenic progenitor /satellite cells reveals distinct molecular signatures for non-dividing myogenic progenitors in activated (*CD44*[High]/*CD98*[+]/*MYOD1*[+]) and dormant

($PAX7^{High}$/$FBN1^{High}$/$SPRY1^{High}$) states. Our approach provides a robust 3D in vitro developmental system for investigating muscle tissue morphogenesis and homeostasis.

## eLife assessment

The authors develop a cell culture system for studies of muscle tissue development and homeostasis. They **convincingly** validate a novel 3D cell model. Their thorough molecular and functional characterization will make this **useful** for future workers in the field.

## Introduction

Novel skeletal muscle model systems are required to further elucidate the process of human myogenesis as well as investigate muscular disorders and potential gene, cell, or drug therapies. Two-dimensional (2D) culture conditions guide pluripotent stem cell (PSC) differentiation toward skeletal muscle lineage using sequential growth factor applications and/or conditional PAX7 expression (*Chal et al., 2015*; *Xi et al., 2017*; *Shelton et al., 2014*; *Borchin et al., 2013*; *Darabi et al., 2012*). Further, surface marker expression can be utilized to isolate myogenic progenitors with in vivo repopulation potential (*Magli et al., 2017*; *Hicks et al., 2018*; *Al Tanoury et al., 2020*; *Sun et al., 2022*). While the few described three-dimensional (3D) differentiation approaches have provided cohorts of terminally differentiated myofibers, focus on potential interactions with the vasculature and nervous system has neglected assessment of the developmental identity or sustainability of myogenic progenitors (*Faustino Martins et al., 2020*; *Maffioletti et al., 2018*; *Rao et al., 2018*). Single-cell technologies increasingly provide databases for deciphering myogenic trajectories and expression profiles of myogenic stem and progenitor cells (*Barruet et al., 2020*; *Rubenstein et al., 2020*; *Xi et al., 2020*), enabling full evaluation of the ability of PSC differentiation protocols to mimic human development. Translation to model muscular dystrophies and investigate potential interventions in vitro necessitates methods that provide expandable populations of muscle progenitors while promoting self-renewal and preserving a quiescent, non-dividing, state (*Quarta et al., 2016*; *Montarras et al., 2005*).

Most vertebrate skeletal muscle progenitors develop from the paraxial mesoderm via transient embryonic developmental structures (somites and dermomyotome) into the skeletal muscle system that spans the whole body. Here, we evaluate human skeletal muscle organoids as a novel system to structurally investigate myogenic differentiation from human induced pluripotent stem cells (iPSCs) in a 3D environment, mimicking pathways described for chicken and mouse (*Buckingham and Rigby, 2014*). We develop a comprehensive supplementation/reduction protocol to first drive differentiation toward paraxial mesoderm through application of the GSK3 inhibitor CHIR99021, BMP inhibitor LDN193189, and bFGF. Subsequent stimulation with WNT1A, SHH, FGF, and HGF is designed to promote derivation of organized structures reassembling neural plate border and dermomyotome. We then aim to arrest neural lineage via FGF removal, while stimulating with HGF to selectively promote propagation of undifferentiated myogenic progenitors and consequent generation of fetal myofibers. Our goal is to provide PAX7+ myogenic progenitors in a non-dividing quiescent state sustainable over weeks after differentiation induction. Single-cell analysis will position cells along the quiescent-activation myogenic trajectory discriminating dormant (PAX7+, FBN1+, SPRY11+, CHODL1+) and activated (CD44+, CD98+, MYOD1+, VEGFA+) states. We thus seek to develop and validate a new skeletal muscle organoid system for investigating human myogenesis with translational potential for disease modeling and therapy development.

## Results

### Generation of human fetal skeletal muscle organoids

Organoid cultures from PSCs often require pre-patterning before Matrigel embedding to promote structural development (*Lancaster et al., 2013*; *Spence et al., 2011*; *Koehler et al., 2017*). In our 3D approach, we provided cells with immediate matrix support upon embryoid body formation to preserve cell state transitions. Initial pre-embedding screening indicated high expression of pluripotent markers *OCT4*, *SOX2,* and *NANOG*, moderate expression of neural tube marker PAX6, and low

**eLife digest** Humans contains around 650 skeletal muscles which allow the body to move around and maintain its posture. Skeletal muscles are made up of individual cells that bundle together into highly organized structures. If this group of muscles fail to develop correctly in the embryo and/or fetus, this can lead to muscular disorders that can make it painful and difficult to move.

One way to better understand how skeletal muscles are formed, and how this process can go wrong, is to grow them in the laboratory. This can be achieved using induced pluripotent stem cells (iPSCs), human adult cells that have been 'reprogrammed' to behave like cells in the embryo that can develop in to almost any cell in the body. The iPSCs can then be converted into specific cell types in the laboratory, including the cells that make up skeletal muscle.

Here, Mavrommatis et al. created a protocol for developing iPSCs into three-dimensional organoids which resemble how cells of the skeletal muscle look and arrange themselves in the fetus. To form the skeletal muscle organoid, Mavrommatis et al. treated iPSCs that were growing in a three-dimensional environment with various factors that are found early on in development. This caused the iPSCs to organize themselves in to embryonic and fetal structures that will eventually give rise to the parts of the body that contain skeletal muscle, such as the limbs. Within the organoid were cells that produced Pax7, a protein commonly found in myogenic progenitors that specifically mature into skeletal muscle cells in the fetus. Pax 7 is also present in 'satellite cells' that help to regrow damaged skeletal muscle in adults. Indeed, Mavrommatis et al. found that the myogenic progenitors produced by the organoid were able to regenerate muscle when transplanted in to adult mice.

These findings suggest that this organoid protocol can generate cells that will give rise to skeletal muscle. In the future, these lab-grown progenitors could potentially be created from cells isolated from patients and used to repair muscle injuries. The organoid model could also provide new insights in to how skeletal muscles develop in the fetus, and how genetic mutations linked with muscular disorders disrupt this process.

expression of mesodermal markers *BRACHYURY* and *MSGN1* (**Figure 1**, **Figure 1—figure supplement 1A**). Upon Matrigel embedding, stimulation with CHIR99021, LDN193189, and bFGF promoted paraxial mesoderm formation through derivation of BRACHYURY⁺ and TBX6⁺ cells (**Figure 1A and C**, **Figure 1—figure supplement 1A**). Immunostaining on day 5 depicted the presence of mesodermal (BRACHYURY⁺), paraxial mesodermal (TBX6⁺), and neuromesodermal (SOX2⁺/BRACHYURY⁺) progenitors (**Gouti et al., 2017**; **Henrique et al., 2015**; **Figure 1C**). From this stage, we attempted to mimic determination front formation and promote anterior somitic mesoderm (ASM) (**Aulehla and Pourquié, 2010**; **Shimozono et al., 2013**) by maintaining constant CHIR99021 and LDN193189 levels, reducing bFGF levels to 50% and simultaneously introducing retinoic acid to the culture. Consequently, distinct PAX3⁺ but SOX2⁻ cells emerged on the organoid surface (**Figure 1D**), followed by a significant downregulation of PSM markers *HES7*, *TBX6*, and *MSGN1* (**Figure 1E**, **Figure 1—figure supplement 1A**). Concomitantly, we observed significant upregulation of ASM marker *MEOX2* and neural crest marker *TFAP2A* but not of *SOX2* or *PAX6*, excluding a shift toward neural tube formation (**Figure 1E and F**, **Figure 1—figure supplement 1A**). Up to this stage, inter-organoid sizes showed small variability (**Figure 1B**). Dermomyotomal fate was promoted by Sonic Hedgehog (SHH) and WNT1A stimulation, while maintaining BMP inhibition avoided lateral mesoderm formation (**Figure 1A**). Expression profiling on day 11 depicted presence of neural tube/crest (*SOX2*, *PAX6*, *TFAP2A*, *SOX10* upregulation) and mesodermal (*UNCX*, *TBX18*, *PAX3*) lineages and further downregulation of PSM markers (**Figure 1E and F**, **Figure 1—figure supplement 1A**). Notably, the dermomyotomal/neural crest marker *PAX7* emerged together with markers that define the dorsomedial (*EN1*) or ventrolateral portion (*SIM1*) of the dermomyotome (**Cheng et al., 2004**; **Figure 1D and F**, **Figure 1—figure supplement 1A**).

Consequently, for favoring myogenesis we stimulated organoid culture with FGF and HGF (**Chargé and Rudnicki, 2004**; **Figure 1A**). Surprisingly, on day 17, organoids constituted a mosaic of neural crest and myogenic progenitor cells. Cells with epithelial morphology were TFAP2A⁺, SOX2⁺, PAX3⁺, and PAX7⁺, indicating formation of neural plate border epithelium (**Roellig et al., 2017**; **Figure 1G**). In cell clusters with mesenchymal morphology, we detected specified migrating PAX3⁺/SOX10⁺ neural crest progenitors and PAX3⁺/SOX2⁻/SOX10⁻ cells of myogenic origin (**Figure 1G**). At this stage,

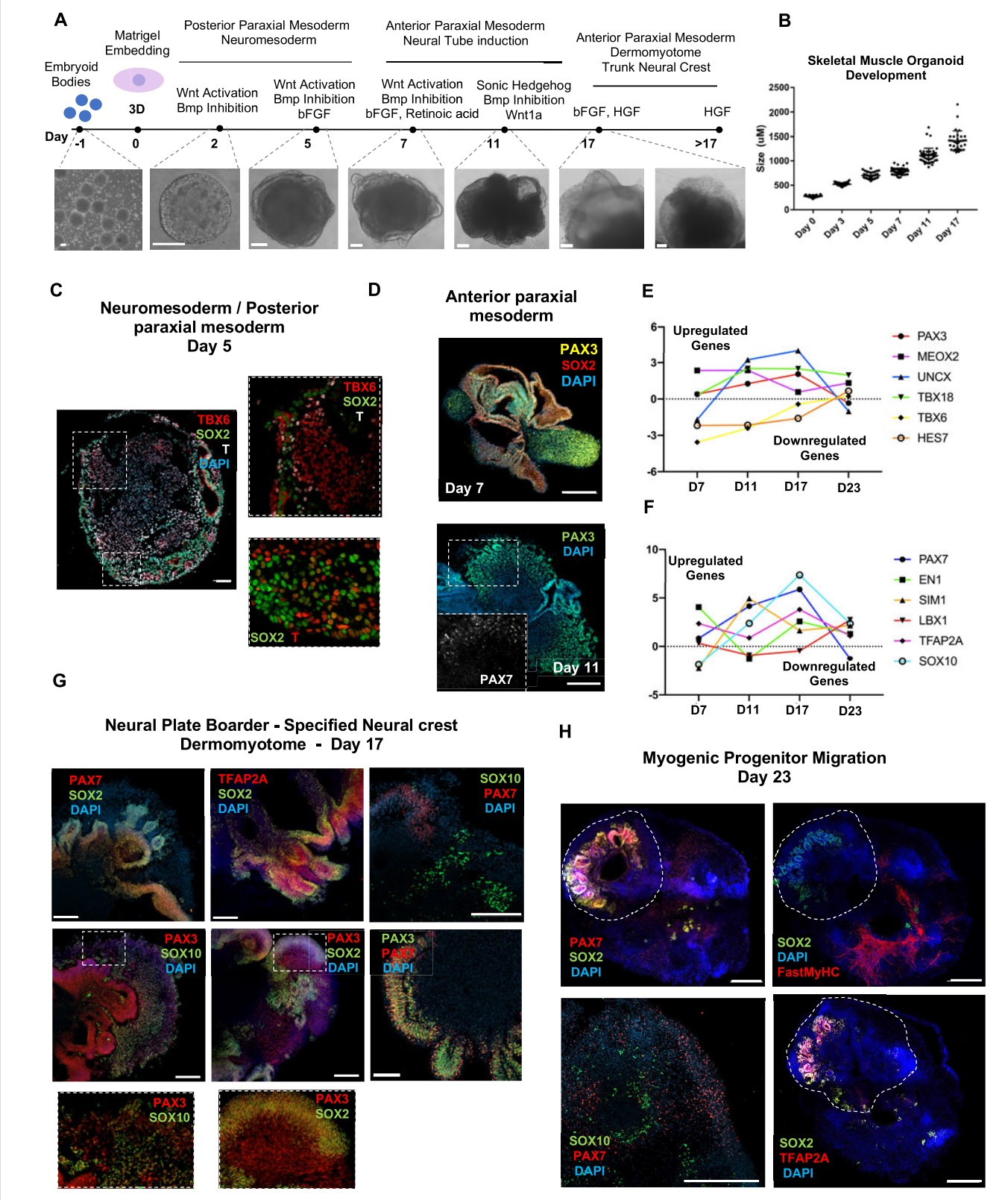

**Figure 1.** Skeletal muscle organoid protocol and correlation to fetal development. (**A**) Brightfield images of myogenic development stages, with corresponding cytokines/growth factors protocol applications. (**B**) Graph depicting organoid development in size (day 0, n = 51; day 3, n = 46; day 5, n = 48; day 7, n = 50; day 11, n = 50; day 17, n = 33; for each timepoint organoids from three independent derivations were measured). (**C**) Representative mesodermal T+, neuromesodermal T+, SOX2+, and paraxial mesodermal TBX6+ organoid expression (in live DAPI- cells) on day 5. (**D**) Representative

*Figure 1 continued on next page*

*Figure 1 continued*

organoid PAX3+, mesodermal SOX2-, and neural SOX2+ expression on day 7; PAX3/PAX7 coexpression on day 11. (**E**) Graph depicting qPCR values for anterior *PAX3/UNCX/MEOX2/TBX18* and posterior *TBX6/HES7* somitic mesodermal markers. (**F**) Graph depicting qPCR values for epaxial and hypaxial dermomyotomal *PAX7/EN1/SIM1/LBX1* and neural crest *TFAP2A/SOX10* markers. (**G**) Representative organoid neural plate border epithelial PAX3+/PAX7+/SOX2+/TFAP2A+, paraxial mesodermal PAX3+/SOX2-, and delaminating specified neural crest progenitor PAX3+/SOX10+ expression on day 17. (**H**) Representative organoid myogenic FastMyHC+, PAX7+, and neural SOX2+/TFAP2A+/SOX10+ expression on day 23. Dashed line represents the location of embryoid body embedded into Matrigel. Statistics: values at each timepoint represent the difference in mean relative expression for each gene (D7 = day 5–day 7, D11 = day 7–day 11, D17 = day 11–day 17, D23 = day 17–day 23) as derived by performing ordinary one-way ANOVA and Tukey's multiple-comparison tests (**E, F**). Scale bars: 200 µm (**G**), 100 µm (**A, D, H**), 50 µm.

The online version of this article includes the following figure supplement(s) for figure 1:

**Figure supplement 1.** Lineage representation and organoid culture progression at early stages of differentiation protocol.

myogenic lineages appeared to be primarily represented by PAX3$^+$ (9.35 ± 0.07%) rather than PAX7$^+$ cells, as the PAX7 expression pattern (15.3 ± 0.1% PAX3$^+$/PAX7$^+$, 5.11 ± 0.13% PAX7$^+$) predominantly overlapped with that of SOX2 and TFAP2A (*Figure 1G* and *Figure 1—figure supplement 1B*). Morphologically, neural plate border and dermomyotomal populations exhibited uneven distribution, and thereby subsequent neural crest and myogenic migration processes introduced organoid size variability (*Figure 1B and G*). From day 17 onward, we omitted FGF stimulation to cease neural crest development and promote delamination/migration of PAX3$^+$/PAX7$^+$/SOX10$^-$ progenitor cells (*Murphy et al., 1994*; *Figure 2A*). Strikingly, until day 23, we observed committed myogenic populations through detection of fast MyHC myofibers in the proximity of PAX7$^+$ but SOX2$^-$/TFAP2A$^-$/SOX10$^-$ negative cells (*Figure 1H*, *Figure 1—figure supplement 1C–E*). Consistently, expression profiling indicated downregulation of neural tube/crest lineage markers and significant upregulation of muscle precursor migrating markers (*Buckingham and Rigby, 2014*), such as LBX1, CXCR4, and MEOX2 (*Figures 1E and 2A*, *Figure 2—figure supplement 1A* ).

At 8 wk, organoids showed profound changes in transcription profiling (*Figure 2*, *Figure 2—figure supplement 1B*). Gene ontology enrichment analysis highlighted an ongoing developmental transition with muscle development, muscle contraction, and myofibril assembly among the top upregulated gene ontology terms and neurogenesis and nervous system development among the top downregulated terms (*Figure 2B*). In addition, we detected downregulation of key markers characterizing neural tube and neural crest development (*Soldatov et al., 2019*), such as *PAK3*, *DLX5*, *B3GAT1*, *FOXB1*, *HES5,* and *OLIG3* (*Figure 2*, *Figure 2—figure supplement 1C*). Interestingly, we could spatially visualize this process using immunocytochemistry for SOX2, TFAP2A, and SOX10-expressing cells that were restricted to the inner portion of the organoid, and probably not susceptible to culture stimulation at 5 wk (*Figures 1H, 2C and D* , *Figure 2—figure supplement 1*). This neural/myogenic lineage spatial orientation could be visualized even on day 84 through the presence of TUJ1$^+$ neurons confined to inner organoid areas and close to SOX2$^+$ epithelium, while FastMyHC$^+$ myofibers occupied exterior areas (*Figure 2E*). On the other hand, substantial proportions of migratory cells in proximity of FastMyHC$^+$, MF20$^+$ myofibers expressed PAX7 (40 ± 0.3%) but not SOX2, TFAP2A, SOX10, or MYOD1 (3.55 ± 0.32%) (*Figure 2F*, *Figure 2—figure supplement 1D and E*). This behavior is further illustrated by the presence of MYOD1$^+$ cells confined to organoid periphery (*Figure 2D*).

Culture progression led to a significant increase in PAX7$^+$ and MYOD1$^+$ cells at 8 wk (45.3 ± 3,4% and 49.8 ± 0.62%, respectively) (*Figure 2F*, *Figure 2—figure supplement 1E*), suggesting an interval in which organoid culture predominantly commits to myogenic lineage. Expression profiling at this stage depicted upregulation of markers characteristic for limb migrating myogenic cells, for example, *LBX1*, *PAX7*, *SIX1/4*, *EYA1/4*, *PITX2*, *MYF5*, *TCF4*, and *MSX1* (*Figure 2*, *Figure 2—figure supplement 1A*). Consequently, HOX gene cluster profiling at 4 wk correlated *HOX A9*, *B9*, *C9* upregulation with limb bud site, while upregulation of the *HOX* 10–13 cluster during 8–16 wk attributed later organoid development to a more distal limb axial anatomical identity (*Shubin et al., 1997*; *Xu and Wellik, 2011*; *Raines et al., 2015*; *Figure 2G*).

## Lineage representation and developmental identity for skeletal muscle organoids

Analysis of organoid culture at single-cell resolution indicated predominant presence of skeletal muscle lineage (n = 3945 cells, 91.26% of total population), complemented with two smaller cell clusters of

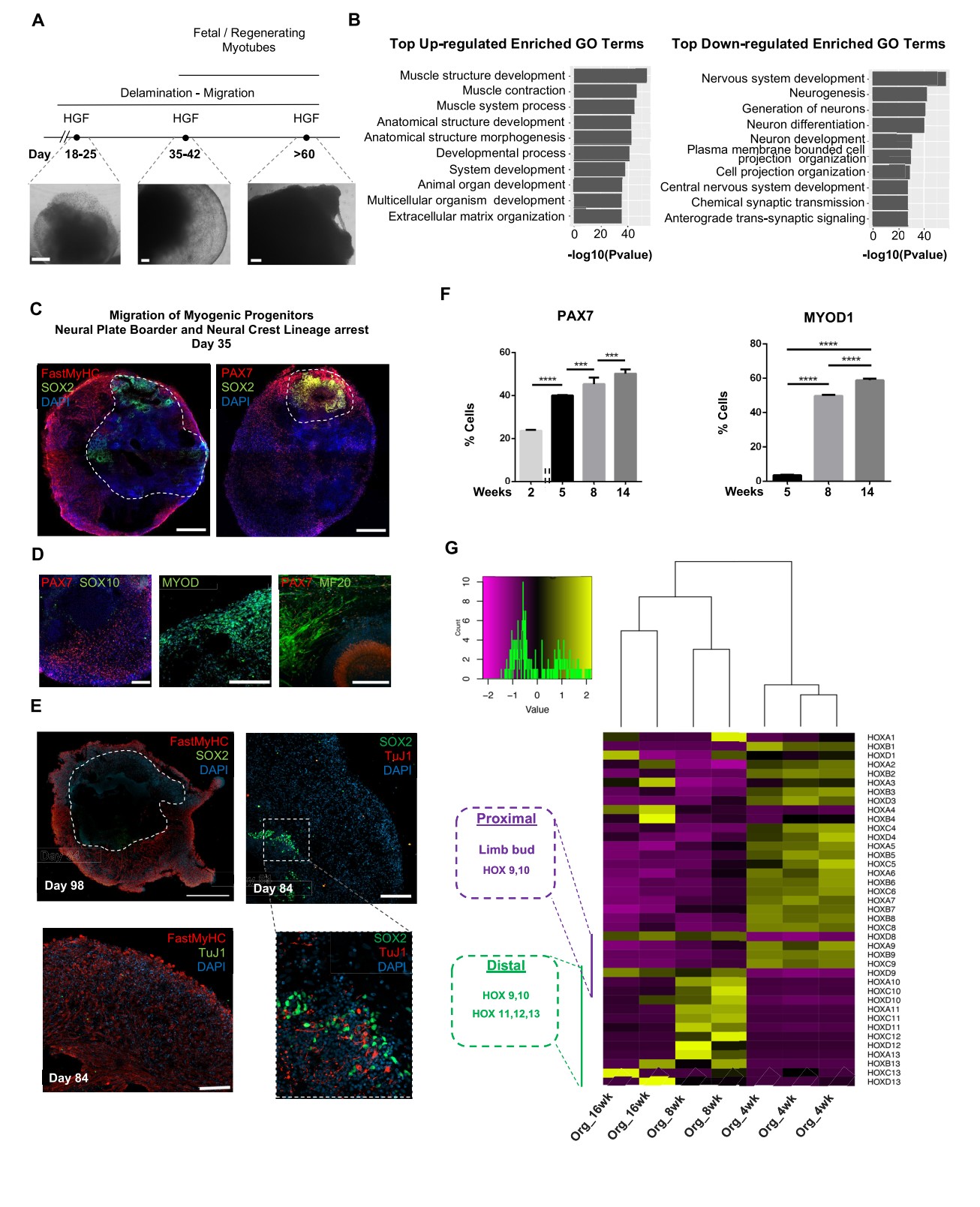

**Figure 2.** Neural lineage development during skeletal muscle organoid progression. (**A**) Stepwise brightfield images depicting delamination/migration of progenitor population during organoid culture progression and corresponding myofiber formation. (**B**) Gene ontology enrichment analysis comparing 4 and 8 wk organoids attributes muscle identity at 8 wk post differentiation and highlights muscle system development and neural lineage arrest among the top upregulated and downregulated gene ontology terms, respectively. (**C–E**) Organoid overview on day 35 indicates predominant

*Figure 2 continued on next page*

*Figure 2 continued*

expression of FastMyHC[+] and PAX7[+] myogenic populations, while SOX2[+] neural populations demarcate SOX2 neural plate border epithelium location as observed at earlier stages (day 16) (**C**); PAX7 cells are of myogenic origin (PAX7[+]/SOX10[-]), MF20[+] myotubes are in their proximity and MYOD1[+] cells appear at organoid periphery (**D**); TUJ1[+] neurons are restricted to inner organoid areas and close to SOX2[+] epithelium, while FastMyHC[+] myofibers occupy exterior organoid areas (**E**). (**F**) Histographs based on FACS intracellular quantification depicting percentage of PAX7[+] or MYOD1[+] cells through differentiation protocol. For each replicate, 10 organoids were pooled, n = 10. Statistics: *p<0.05, **p<0.01, ***p<0.001, ****p<0.0001, ns: not significant. (**G**) Heatmap of *HOX* gene cluster emphasizes organoid culture limb axial anatomical identity by depicting transition from an initial limb bud (*HOX 9–10*) toward a more distal identity (*HOX 11–13*) at 8 and 16 wk post differentiation, respectively. Scale bars, 500 µm (**C**), 200 µm (**A, D, E**).

The online version of this article includes the following figure supplement(s) for figure 2:

**Figure supplement 1.** Myogenic versus neural fate during organoid development.

mesenchymal/'fibroadipogenic' (n = 165 cells, 3.82% of total population), and neural (n = 213 cells, 4.93% of total population) origin (***Figure 3A***, ***Figure 3—figure supplement 1A***). Skeletal muscle lineages further separated into distinct subclusters: myogenic muscle progenitors in non-dividing (*PAX7[+]/PAX3[-]*) and mitotic states (*PAX7[+]/CDK1[+]/KI67[+]*), myoblasts (*MYOD1[+]*), myocytes (*MYOG[+]*), and myotubes (*MYH3[+]*) (***Figure 3A***, ***Figure 3—figure supplement 1A and B***). Further, investigation of mesenchymal cluster identity suggested fibroadipogenic potential of a distinct cell population with *PDGFRa* among the top upregulated genes (***Uezumi et al., 2010***; ***Uezumi et al., 2011***; ***Xi et al., 2020***; ***Figure 3A and F***, ***Figure 3—figure supplement 1A and B***). Consistently, we detected upregulation of fibrotic markers. Moreover, adipogenic potential was highlighted through upregulation of *PREF-1* and *EBF2* (***Figure 3F***). Occasionally, we detected structures resembling adipocytes from 9–10 wk onward, with adipogenic nature verified by PRDM16 and Oil red O positive staining (***Figure 3B and D***, ***Figure 3—figure supplement 1C and D***). A SOX2[+]/TUBB3[+] positive neural cluster potentially derived from the neural plate border epithelium at younger stages could be located toward organoid interior (***Figure 3C***, ***Figure 3—figure supplement 1E***).

Myogenesis progression based on *t*-SNE feature and Violin plots of key markers from each stage indicated gradual transitions from myogenic progenitor to myotube subclusters (***Figure 3E***, ***Figure 3—figure supplement 1F***). MyHC expression profiling at single-cell resolution demonstrated predominant presence of embryonic (*MYH3[+]*) and perinatal (*MYH8[+]*) myofibers, co-expressing β-Enolase and M-cadherin in the vicinity of PAX7[+] cells (***Figure 3G–J***). Consistently, we detected MCAM expression on muscle subclusters (***Figure 3G***), while bulk RNA-seq at 16 wk highlighted expression of additional *MyHC* isoforms, for example, *MYH2 (IIa)*, *MYH4(IIb)* and transcription factor NFIX (***Moore and Walsh, 1993***; ***Barbieri et al., 1990***; ***Alexander et al., 2016***; ***Messina et al., 2010***). At more mature stages, we detected adult *MYH1* or slow *MYH7* isoforms (***Figure 3***, ***Figure 3—figure supplement 1G***), presumably due to myofiber stimulation via spontaneous contraction. In agreement, differential expression comparison between 8 and 16 wk organoids indicated less variance in transcription profiling (***Figure 3***, ***Figure 3—figure supplement 1H***, ***Figure 4A***). Consistently, sarcomere organization, ion transport, response to stimulus, and synapse structure maintenance were among the upregulated gene ontology terms and mitosis, cell cycle, DNA packaging, and nuclear division among the top downregulated terms (***Figure 3***, ***Figure 3—figure supplement 2B–D***). Notably, ongoing maturation did not affect pool of progenitor cells, as even at 14 wk we could report significant increases in PAX7[+] (50,16 ± 2.19%) and MYOD1[+] (58.53 ± 0.92%) cells (***Figures 2F and 3***, ***Figure 3—figure supplement 2E and F***).

## Functionality and maturation of organoid-derived myofibers

Regarding localization and functionality of organoid-derived myofibers, immunocytochemistry revealed positive staining for dystrophin and a continuous laminin sheath around individual muscle fibers. Ultrastructure and two-photon microscopy analysis depicted well-developed sarcomeres (***Figure 3***, ***Figure 3—figure supplement 3A–D***). Moreover, patch-clamp experiments, upon superfusion of cells with acetylcholine (ACh), indicated inward current generations that rapidly declined to a low steady-state level (***Figure 3K***, ***Figure 3—figure supplement 3E***). The I/V curve of this current component showed almost linear behavior and reversal potential around 0 mV (***Figure 3L***). These data are in line with previous studies that described currents with analogous properties as characteristic nAChR currents (***Jahn et al., 2001***; ***Shao et al., 1998***). Application of a fluorescent biosensor for monitoring changes in cytosolic $Ca^{2+}$ (Twitch2B) revealed nAChR efficiency in modulating intracellular

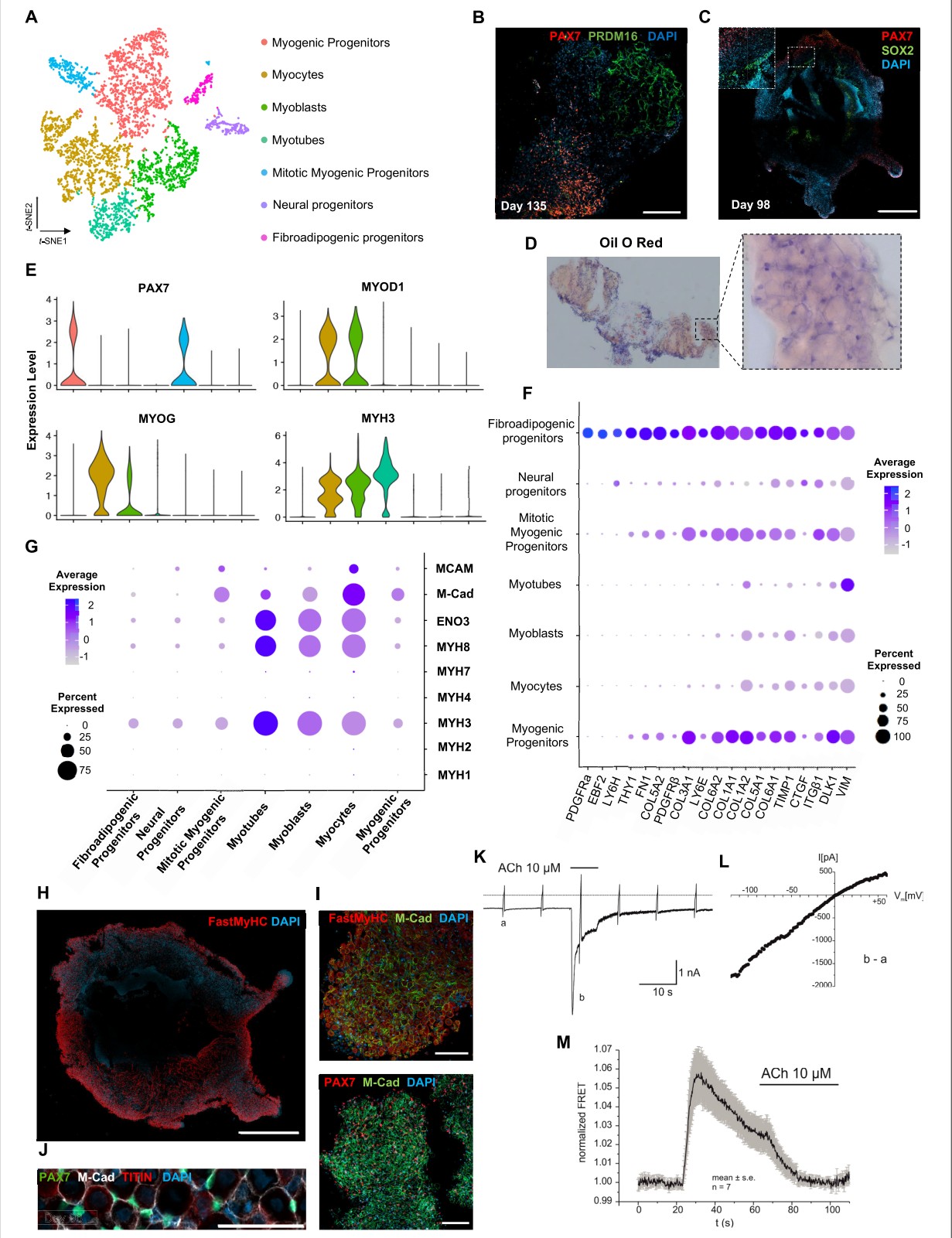

**Figure 3.** Skeletal muscle organoid characterization at single-cell resolution. (**A**) *t*-SNE visualization of color-coded clustering (n = 4323 cells) at 12 wk post differentiation highlights the predominant presence of skeletal muscle lineage, represented by clusters corresponding to myogenic progenitors (n = 1625 cells, 37% of total population) in non-dividing (n = 1317 cells) and mitotic (n = 308 cells) state, myoblasts (n = 731 cells), myocytes (n = 1147 cells), and myotubes (n = 442). Additionally, mesenchymal and neural lineages are represented by two smaller clusters of fibroadipogenic (n = 165 cells)

*Figure 3 continued on next page*

*Figure 3 continued*

and neural (n = 213 cells) progenitors, respectively. (**B**) Immunocytochemistry on day 135 indicates derivation of PRDM16+ adipocyte clusters distinct from PAX7+ myogenic progenitors. (**C**) Organoid overview on day 98 depicts expression of PAX7+ myogenic populations and highlights SOX2+ neural populations toward organoid interior. (**D**) Positive areas with Oil O Red staining indicate derivation of adipocytes in organoid culture on day 135. (**E**) Violin plots of key markers *PAX7, MYOD1, MYOG, MYH3* from each stage as in (**A**) depict relative expression levels and emphasize gradual transition from myogenic progenitor to myotube subcluster. (**F**) Dot plot showing expression of representative genes related to adipogenesis and fibrogenesis across the seven main clusters. Circle area represents the percentage of gene+ cells in a cluster, color reflects average expression level (gray, low expression; blue, high expression). (**G**) Dot plot showing expression of representative genes related to fetal myogenesis across the seven main clusters. Circle area represents the percentage of gene+ cells in a cluster, color reflects average expression level (gray, low expression; blue, high expression). (**H–J**) Representative organoid overview on day 98 indicates predominant expression of Fast MyHC fetal myofibers (**H, J**), positive for M-Cadherin and in PAX7+ cells proximity (**I, J**). (**K**) Representative recording (n = 6) of acetylcholine (ACh)-induced changes in holding current in a single skeletal muscle cell. ACh (10 μM) was applied as indicated by the bar. Holding potential –90 mV. Downward deflections represent membrane currents in response to depolarizing voltage ramps (duration 500 ms) from –120 mV to +60 mV. Dashed line indicates zero current level. (**L**) I/V curve of nAChR currents obtained by subtraction of voltage ramp-induced changes of current in the presence and absence of ACh (10–5 mol/l), corresponding lowercase letters b – a. (**M**) Summarized FRET recordings from skeletal muscle cells transfected with Twitch2B to monitor the increase in [Ca2+]i during ACh application (h). Scale bars, 1 mm (**C, H**), 200 μm (**B**), 100 μm (**I, J**).

The online version of this article includes the following figure supplement(s) for figure 3:

**Figure supplement 1.** Single-cell RNA-seq expression profiling and lineage representation in organoid culture at week 12.

**Figure supplement 2.** Skeletal muscle organoid culture maturation and identity.

**Figure supplement 3.** Functional properties of organoid-derived skeletal muscle myofibers.

[Ca$^{2+}$]. Summarized FRET recordings (*Figure 3M*, *Figure 3—figure supplement 3F*), following application of ACh (10 μM), illustrated a rapid increase in FRET ratio that gradually declined in the presence of ACh, probably reflecting desensitization of nAChRs. These results demonstrated that nAChR in skeletal muscle cells are functional in terms of inducing Ca$^{2+}$ release from intracellular stores.

Using our organoid protocol, we successfully derived fetal muscle progenitors and electrophysiologically functional myofibers from hiPSC lines with wild type and Duchenne muscular dystrophy genetic backgrounds (*Figure 3*, *Figure 3—figure supplement 3E and F*).

## Identity and sustainability of organoid-derived PAX7 myogenic progenitors

Skeletal muscle organoids remarkably foster sustainable propagation of PAX7+ progenitors (*Figures 2F and 3A*). Initial screening of PAX7+ progenitors indicated expression of several satellite cell markers (*Fukada et al., 2007*), such as *CD82, CAV1, FGFR1, FGFR4, EGFR, M-Cadherin, NCAM, FZD7, CXCR4, ITGβ1, ITGA7, SCD2,* and *SCD4* (*Figure 4*, *Figure 4—figure supplement 1A*). Further investigation verified NOTCH signaling activity (*HES1, HEYL, HEY1,* and *NRARP*) in the myogenic subcluster, while dormant myogenic progenitors exhibited high expression of *SPRY1* and cell cycle inhibitors *p57 (CDKN1C), p21,* and *PMP22* (*Xi et al., 2020*; *Fukada et al., 2007*; *Bjornson et al., 2012*; *Shea et al., 2010*; *Figure 4*, *Figure 4—figure supplement 1B*, *Figure 4—figure supplement 2A and D*). Moreover, proliferation assays performed at the same stage (14 wk) demonstrated 4.78 ± 0.28% edU+ cells, while substantial proportions of PAX7+ myogenic progenitors remained quiescent (*Figure 4*, *Figure 4—figure supplement 1C and D*). *t*-SNE clustering divided myogenic progenitors into three distinct groups with unique molecular signatures: '*CD44$^{high}$*,' '*FBN1$^{high}$*,' and '*CDK1$^+$*' clusters (*Figure 4A*, *Figure 4—figure supplement 2A*). The '*CD44high*' cluster, further characterized by *CD98* upregulation, adopted a molecular signature similar to activated satellite cell state (*Porpiglia et al., 2017*; *Figure 4A and B*, *Figure 4—figure supplement 2A and D*). In this state, myogenic progenitors expressed *VEGFA* (*Figure 4—figure supplement 2A and D*) as previously described for murine satellite cells (*Verma et al., 2018*). Consistently, the CD44+/PAX7+ 'activated' myogenic population located at sites more accessible to HGF signaling, for example, exterior organoid areas and forming bulges (*Figure 4B*). The '*FBN1$^{High}$*' subcluster was further characterized by '*PAX7$^{high}$/SPRY1$^{high}$/CHODL$^{high}$/FBN1$^{high}$*' expression (*Figure 4A*, *Figure 4—figure supplement 2A and D*). *PAX7high* cells within the dormant state co-expressed *NOTCH3, JAG1,* and *LNFG,* together with *CHODL* markers. Further, we verified the presence of FBN1+ microfibrils, which, compared to CD44+ cells, occupied areas without direct access to activation signals (*Figure 4C*). Lastly, the '*CDK1$^+$*' cluster was the only proliferative population, further expressing markers mediating satellite cell activation and proliferation, such as

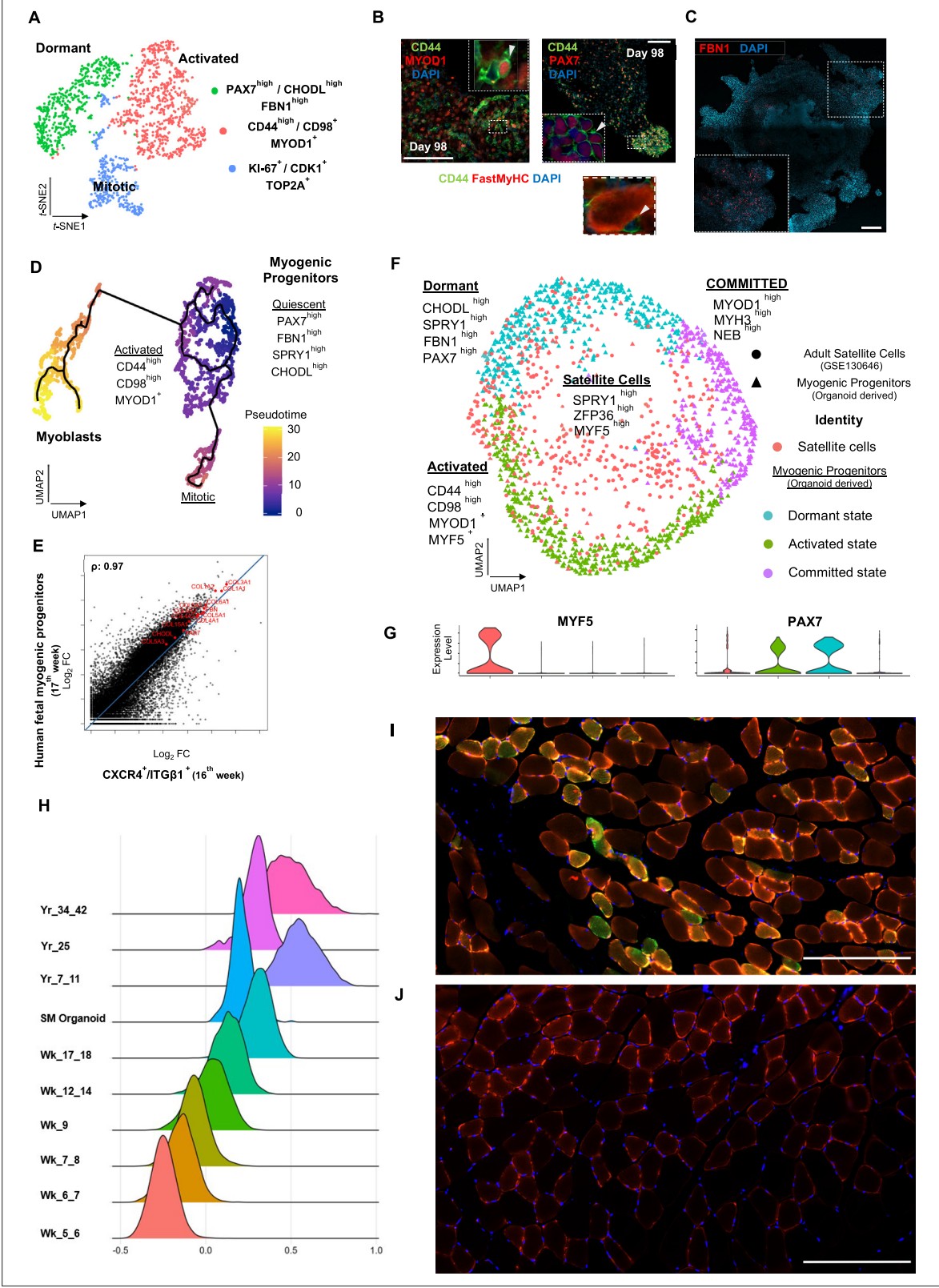

**Figure 4.** Myogenic progenitor identity and comparison to progenitors derived from fetal and adult muscle tissue. (**A**) *t*-SNE plot visualization of color-coded clustering indicates myogenic progenitor subcluster with distinct molecular signatures: 'dormant' *PAX7high*/*CHODLhigh*/*FBN1high*, 'activated' *CD44high*/*CD98+*/*MYOD1+*, and 'mitotic' *KI-67+*/*CDK1+*/*TOP2A+*. (**B**) Organoid overviews on day 98 illustrate CD44 and PAX7-expressing cells at developing sites, which are more accessible to HGF activation signal, specificity of CD44 on MYOD+/PAX7+ progenitor-expressing cells (arrows) and

*Figure 4 continued on next page*

*Figure 4 continued*

absence from FastMyHC⁺ positive myofibers is highlighted. (**C**) FBN1⁺ microfibrils are located toward organoid interior. (**D**) Pseudotime ordering for myogenic progenitors and myoblast corresponding clusters highlights distinct developmental trajectories promoting myogenic commitment and self-renewal. (**E**) Correlation coefficient plot for Log2 fold change (Log2 FC) values for isolated myogenic progenitors from human fetal tissue (17w) and FACS-sorted CXCR4⁺/ITGβ1⁺ organoid-derived myogenic progenitors (16w). *PAX7, COL1A1, COL3A1, COL4A1, COL5A1, COL15A1, FBN1,* and *CHODL* and further extracellular matrix-related genes are highlighted on the plot. Pearson's correlation coefficient, rho = 0.9 for global expression comparison and rho = 0.97 for selected genes. (**F**) UMAP color-based clustering divides non-dividing myogenic progenitors and adult satellite cells into four clusters with distinct molecular signatures: satellite cells characterized by *SPRY1^high^/ZFP36^high^/MYF5^high^* expression, co-clustered with dormant *SPRY1^high^/FBN1^high^/CHODL^high^/PAX7^high^*, activated *CD44^high^/CD98^high^/MYOD1⁺*, and committed, *NEB ^high^/MYH3^high^/MYOD1^high^* organoid-derived myogenic progenitors. Dots correspond to adult satellite cells from GSE130646 database, triangles correspond to organoid-derived myogenic progenitors. (**G**) Violin plots depicting relative expression levels of markers specific for quiescent PAX7 or activated MYF5 muscle stem cell state across adult satellite cells (GSE130646) and organoid-derived myogenic progenitors subclusters. (**H**) Ridge plots of developmental score distribution of myogenic progenitors across in vivo or in vitro stages based on the difference between upregulated satellite cell and embryonic markers from human reference atlases for weeks (Wk) 5–18 embryonic and fetal stages, years (Yr) 7–42 adult satellite cells and skeletal muscle (SM) organoid. (**I, J**) In vivo engraftment potential of human myogenic progenitors. 100.000 CD82+ sorted human cells were injected into *Tibialis anterior* muscle of nude mice. (**I**) Control mice were not injected (**J**). Six weeks post transplantation, transverse cryosections of muscle fibers were stained with huLamin A/C (green), dystrophin (red), and DAPI (blue). Human cells appear green and red in contrast to murine cells, which only show a dystrophin-positive red staining. Scale bars 200 μm in (**I, J**).

The online version of this article includes the following figure supplement(s) for figure 4:

**Figure supplement 1.** Subclustering of myogenic progenitors and NOTCH signaling.

**Figure supplement 2.** Pseudotime ordering of myogenic progenitor revealing distinct states and cell fate decisions.

**Figure supplement 3.** Organoid-derived myogenic progenitors and correlation to fetal muscle progenitors.

**Figure supplement 4.** Organoid-derived myogenic progenitors and correlation to adult human satellite cells.

**Figure supplement 5.** Characterization of cell–cell communication network for all clusters at week 12 of human skeletal muscle organoid development.

**Figure supplement 6.** Reproducibility of organoid culture at early and mature stages.

**Figure supplement 7.** Comparison between 2D in vitro myogenic differentiation protocols and in vivo staging.

*DEK* and *EZH2* (*Cheung et al., 2012*; *Juan et al., 2011*; *Figure 4A and D*, *Figure 4—figure supplement 1A*, *Figure 4—figure supplement 2A and B*).

Pseudotemporal ordering of myogenic progenitors indicated that the '*FBN1^High^*' subcluster was the main progenitor population residing in dormant state, which by downregulating *PAX7, FBN1,* and *CHODL* and upregulating *CD44, MYOD1, CD98 (SLC3A2),* and *VEGF* generated activated state (*Figure 4D*, *Figure 4—figure supplement 2B*). In activated state, myogenic progenitors that upregulate *MYOD1* entered mitosis, proliferated, and followed a trajectory leading to myogenic commitment (*Figure 4D*, *Figure 4—figure supplement 2B*). This commitment was further accompanied by *PARD3, p38a MAPK,* and *CD9* expression (*Porpiglia et al., 2017*; *Troy et al., 2012*; *Figure 4—figure supplement 2E*). Conversely, myogenic progenitors that upregulated *NOTCH3, SPRY1,* and *PAX7* followed a loop trajectory that reinstated dormant stem cell state (*Figure 4—figure supplement 2D and E*). Interestingly, the '*FBN1^high^*' cluster highly expressed extracellular matrix collagens, for example, *COL4A1, COL4A2, COL5A1, COL5A2, COL5A3,* and *COL15A1*. Notably, such expression declined upon commitment and differentiation (*Figure 4—figure supplement 2C and F*).

## Organoid-derived myogenic progenitor comparison to human fetal and adult progenitor/stem cells

To evaluate developmental identity of the myogenic cluster, we isolated ITGβ1⁺/CXCR4⁺ organoid-derived myogenic progenitors via FACS (*Garcia et al., 2018*) and compared to human fetal muscle progenitors, demonstrating high similarity (Pearson's correlation co-efficiency, rho = 0.9), with myogenic markers such as *PAX7, MYF5, MYOG,* and *MYOD1* at comparable expression levels (*Figure 4—figure supplement 3A–F*). Differential expression comparison verified expression of extracellular matrix collagens and proteins, such as *COL4A1, COL5A1, COL6A1, COL15A1, FBN1,* and *CHODL*, in myogenic progenitors similar to 17th week human fetal tissue (Pearson's correlation co-efficiency, rho = 0.97) (*Figure 4E*). Further, to evaluate myogenic potency in vitro, isolated ITGβ1⁺/CXCR4⁺ organoid-derived myogenic progenitor cells were re-plated and allowed to differentiate under the same organoid culture protocol, which demonstrated capacity to generate Fast-MyHC+ and PAX7+ populations within 10 d (*Figure 4—figure supplement 3B and C*). Subsequently,

comparison to available transcriptomic dataset of human adult satellite cells at single-cell resolution divided myogenic progenitors and adult satellite cells into four clusters with distinct molecular signatures (*Figure 4F*). Interestingly, myogenic progenitors were enriched for extracellular matrix proteins, while satellite cells mainly upregulated genes repressing transcription/translation, such as *ZFP36* and *TSC22D1*, or related to early activation response, such as *MYF5, JUN,* and *FOS* (*Figure 4—figure supplement 4A and B*). In line, organoid-derived myogenic progenitors exhibited higher *NOTCH* signaling activity in comparison to satellite cells, with *NOTCH3* and *RBPJ* being among enriched markers (*Figure 4—figure supplement 4B*). In contrast, adult satellite cells exhibited $PAX7^{low}/MYF5^{high}$ expression profiling, presumably due to tissue dissociation, thereby indicating a tendency for activation rather than preservation or reinstatement of quiescency (*Machado et al., 2017*; *Seale et al., 2000*; *Figure 4G*). Pseudotime ordering showed two distinct clusters, with adult satellite cells downstream from non-dividing myogenic progenitors (*Figure 4—figure supplement 4C*). Consistently, downregulation of genes promoting quiescence like *PAX7, NOTCH3,* and *RBP,* and upregulation of activation genes like *MYF5, JUN,* and *FOS* along the trajectory (*Figure 4—figure supplement 4D*), was a further indication that organoid-derived myogenic progenitors resided in dormant non-dividing state and that our organoid platform promoted sustainability of myogenic progenitors.

Cell–cell communication analysis of organoids at 12 wk indicates that myogenic progenitors influence their own fate, mainly with extracellular matrix (ECM)-related signals, as well as receive signals predominantly from the mesenchyme but not the neural progenitor cluster (*Figure 4—figure supplement 5*).

To evaluate for reproducibility of organoid development, we applied diffusion map analysis on qPCR-based expression analysis of 32 genes at early stages and integrative analysis on scRNAseq datasets of mature stages of organoid development. The data indicate highly conserved cluster representation of myogenic progenitors at all states, together with skeletal muscle myofibers, fibroadipogenic progenitors, and neural progenitors-related clusters (*Figure 4—figure supplement 6*).

Organoid-derived myogenic progenitors exhibited expression profiling similar to that of stage 4 myogenic progenitors, correlating to 17th–18th week human fetal development, as reported in the human skeletal muscle atlas (*Xi et al., 2020*). *SPRY1, PLAGL1, POU4F1, TCF4, PAX7, CD82,* and *CD44* markers, as well as ECM proteins, nuclear factor I family members (*NFIA, NFIB, NFIC*), and *NOTCH* signaling, are specifically upregulated markers. Furthermore, in comparison to 2D culture protocols, our organoid approach preserved myogenic progenitor dormancy without activating MYOD1 and exhibited higher expression of ECM proteins and key markers characterizing stage 4 fetal myogenic progenitors (*Figure 4—figure supplement 7*). *ERBB3* and *NGFR* markers (*Hicks et al., 2018*) demarcate populations of earlier developmental stages and were not upregulated either in organoid-derived or stage 4 fetal myogenic progenitors (*Figure 4—figure supplement 7*). In addition, organoid myogenic progenitors and stage 4 fetal myogenic progenitors both downregulated cycling genes like *MKI67, TOP2A, EZH2,* and *PCNA* (*Figure 4—figure supplement 7*).

In addition, we have performed a developmental score distribution of myogenic progenitors based on the difference between upregulated satellite cell and embryonic markers from the human reference myogenic atlases (*Xi et al., 2020*) and adult satellite cell data (*Rubenstein et al., 2020*; *Xi et al., 2020*) in comparison to our organoid protocol (*Figure 4H*). Organoid-derived myogenic progenitors represent the late fetal stage of maturation partially overlapping with adult satellite cell scoring. We note the heterogeneity of adult satellite cell populations when performing developmental scoring in line with recent reports (*Barruet et al., 2020*).

Finally, we transplanted CD82-positive progenitors from our organoids into the *Tibialis anterior* (TA) muscle of immunodeficient mice to complement our study with an in vivo experiment (*Alexander et al., 2016*; *Marg et al., 2019*; *Al Tanoury et al., 2020*). The CD82 positivity used for FACS selection of myogenic progenitors prior to transplantation almost exclusively overlaps with Pax7-positive cells, being a subcluster of them (*Figure 4—figure supplement 4E*). Six weeks post transplantation, we could verify clusters of human Lamin A/C-positive cells in the transplanted but not in the control group (*Figure 4I and J*). We have measured the size of myotubes with 41 ± 6 μm for the human and 63 ± 7 μm for the mice mean diameters (n = 15 each).

# Discussion

Human skeletal muscle organoids offer a new cell culture system to study human myogenesis, in particular fetal myogenic progenitors. We demonstrate that modulation of Matrigel-embedded embryonic bodies with WNT, BMP, and FGF signaling at early stages leads to paraxial mesoderm formation (*Figure 1B*). Further, under guided differentiation, we could promote concomitant development of neural and paraxial mesodermal lineages and derive mesodermal populations with somitic and dermomyotomal origin (*Figure 1C–F*). With WNT1A and SHH stimulation, neural lineage is directed toward dorsal neural tube/crest development which benchmarks the structural recapitulation of neural plate border epithelium (*Figure 1G*). In vitro neural plate border can be utilized to assess transcriptomic networks and cell fate decision during human neural crest formation.

Delaminating from the dermomyotome, undifferentiated PAX3 progenitor cells reorient and align in the cranio-caudal axis to form the underlying myotome or completely detach from the dermomyotome and migrate into the surrounding mesenchyme at the limb sites, where they propagate and become committed to skeletal muscle progenitors (*Relaix et al., 2005*). By stimulating organoid culture at the neural plate border/dermomyotomal stage with bFGF/HGF, we could further visualize both migration of myogenic progenitors and migration/specification of neural crest populations (*Figures 1A, G, H and 2A*). Further, by omitting FGF during organoid development, we could detect a continuous upregulation of genes involved in the myogenic migration process, such as *LBX1*, *PAX3*, *PAX7* and *CXCR4*, but not in genes characterizing neural tube or neural crest development, such as *SOX10*, *TFAP2A*, *PAK3*, *DLX5*, *B3GAT1*, *FOXB1*, *HES5,* and *OLIG3*. This indicates that organoid culture conditions and specifically stimulation with HGF favored skeletal muscle over neural lineage (*Figure 2C–E*). Interestingly, we could show that by stimulating organoid culture with SF/HGF, an essential signal for detachment and subsequent migration, but not for specification of the cells at the lateral lip of the dermomyotome (*Dietrich et al., 1999*), we could preserve the PAX3+/PAX7+ myogenic migratory population in an undifferentiated and uncommitted state (*Figure 2D and E*). Strikingly, expression profiling based on HOX gene cluster supported this notion as over time the organoid culture adopted more distal than proximal limb axial anatomical identity (*Figure 2F*).

Fetal development is characterized by PAX3+ to PAX7+ myogenic progenitor transition (*Seale et al., 2000*; *Relaix et al., 2005*), which we were able to demonstrate in our organoid culture. Our data further support organoid culture representing fetal stages as we could detect NFIX upregulation and verify the presence of myofibers expressing fetal MyHC isoform as well as NCAM, M-Cad, or MCAM in the proximity of PAX7 progenitors (*Figure 3G–J*). Consequently, transcriptome comparison indicated high similarity to human fetal muscle tissue (17 wk, Pearson correlation, rho = 0.9) (*Figure 4E*), as well as expression of several satellite cell markers (*Figure 4—figure supplement 1A*). This was further verified by comparison to a single-cell dataset from the human skeletal muscle cell atlas (*Figure 4*, *Figure 4—figure supplement 4*) Interestingly, single-cell resolution showed adult satellite cell clusters with organoid-derived myogenic progenitors (*Figure 4F*). In addition, pseudotime ordering indicated that organoid-derived myogenic progenitors reside developmentally upstream of activated satellite cells, upregulating markers associated with quiescence such as NOTCH signaling and extracellular matrix proteins (*Figure 4—figure supplement 2E and F*). Preservation of myogenic progenitors in a non-dividing state without activating MYOD1 and self-renewal (*Figure 4D*, *Figure 4—figure supplement 2E and F*) appeared to be responsible for the observed sustainable propagation of PAX7 progenitor cells even 14 wk post differentiation (*Figures 2F and 3A*).

Patterning in our 3D protocol provides progenitors in a more mature late fetal state partially overlapping with adult satellite cell developmental scoring (*Figure 4H*). The 'uncommitted' Pax7 progenitor status is demonstrated by our precursor populations bulkRNAseq and scRNAseq profiling (*Figure 4A, E and F*). In this context, we could observe high expression of extracellular matrix proteins and upregulated NOTCH signaling in dormant non-dividing myogenic progenitors (*Figure 4—figure supplement 2A and E*). This phenotype is similarly described for human fetal tissue myogenic progenitors (*Figure 4E*, Pearson correlation, rho = 0.97, *Figure 4—figure supplement 5*). Studies evaluating engraftment potential of satellite cells and fetal muscle progenitors propose that muscle stem cells in a quiescent non-committed state exhibit enhanced engraftment potential (*Hicks et al., 2018*; *Quarta et al., 2016*; *Montarras et al., 2005*; *Tierney et al., 2016*). Our data demonstrate that upon activation and commitment dormant myogenic progenitors downregulate extracellular matrix proteins and upregulate expression of morphogens/receptors that make them susceptible to signals,

like VEGFA, that communicates with vasculature during tissue reconstitution, or CD9, CD44, CD98 participating in cellular activation (*Porpiglia et al., 2017*; *Figure 2—figure supplement 1B and D*). Cell–cell communication analysis revealed that myogenic progenitors influence their own fate mainly with ECM-related signals (*Figure 4—figure supplement 5*). This finding is in line with the nature of in vivo fetal myogenic progenitors (*Tierney et al., 2016*) but also indicates that investigation at the myogenic progenitor level could provide new insights into ECM-related congenital muscular dystrophies. One example is the Ullrich congenital muscular dystrophy, where the ECM alters the muscle environment with progressive dystrophic changes, fibrosis, and evidence for increased apoptosis (*Bönnemann, 2011*).

CD82+ populations from our organoids engraft in the TA muscle of immunodeficient mice (*Figure 4I and J*). It would be of interest for future studies to investigate whether increased engraftment can be achieved in 3D protocols (*Faustino Martins et al., 2020*; *Shahriyari et al., 2022*; ours) versus 2D patterned progenitor cells and to which degree this is attributed to high expression of extracellular matrix proteins. In particular, high Fibrillin1 expression on dormant non-dividing myogenic progenitors could potentially contribute to avoidance of fibrosis by myogenic progenitors through regulation of TGF-β signaling (*Cohn et al., 2007*).

Different phases of human fetal myogenesis have been modeled with 2D differentiation protocols (*Shelton et al., 2014*; *Chal et al., 2015*; *Xi et al., 2017*). Our 3D differentiation protocol does not go beyond these protocols when it comes to provide matured physiologically responsive skeletal muscle cells, which we illustrate with the electrophysiological recording of organoid-derived cells of different origins (*Figure 3K, L and M*, *Figure 3—figure supplement 3E and F*). Structural distinctions like the posterior paraxial mesoderm on day 5 specified neural crest dermomyotome on day 17, myogenic progenitor migration on day 23, and neural crest lineage arrest on day 35 (*Figures 1C–H and 2C*) cannot be similarly observed in 2D protocols. In addition, our 3D organoid protocol provides myogenic progenitors in dormant and activated states for at least 14 wk in culture. We demonstrate that organoid culture sustains uncommitted MyoD1-negative, Pax7-positive myogenic progenitors as well as fibroadipogenic (PDGFRa+) progenitors, both resembling their fetal counterpart. This supply of muscle progenitors preserved in a quiescent state indicates translative potential of the approach. Future work will elucidate signaling transduction pathways during skeletal muscle organoid development to model and understand human myogenesis in more detail.

## Materials and methods

### hiPSCs culture

hiPSC lines, Cord Blood iPSC (CB CD34+, passages 15–35), Duchenne muscle dystrophy iPSC patient line iPSCORE_65_1 (WiCell, Cat# WB60393, passages 22–30), and DMD_iPS1 (passages 21–30) and BMD_iPS1 (passages 17–25) (Boston Children's Hospital Stem Cell Core Facility), LGMD2A iPSC, and LGMD2A-isogenic iPSC (*Dorn et al., 2015*; *Mavrommatis et al., 2020*; *Panopoulos et al., 2017*; *Park et al., 2008*) were cultured in TESR-E8 (StemCell Technologies) on Matrigel GFR (Corning)-coated 6-well plates. Cells have been tested negative for mycoplasma. No experiments on human participants were performed. The use of reprogrammed human iPSC lines for research was performed after ethical approval from the ethics commission of the Ruhr-University Bochum, Medical Faculty (15-5401, August 2015).

### Human skeletal muscle organoid differentiation protocol

Prior differentiation, undifferentiated human PSCs, 60–70% confluent, were enzymatically detached and dissociated into single cells using TrypLE Select (Thermo Fisher Scientific). Embryoid bodies formed via hanging drop approach with each droplet containing $3–4 × 10^3$ human single PSCs in 20 µl were cultured hanging on TESR-E8 supplemented with polyvinyl alcohol (PVA) at 4 mg/ml (Sigma-Aldrich) and rock inhibitor (Y-27632) at 10 µM (StemCell Technologies) at the lid of Petri dishes. At the beginning of skeletal muscle organoid differentiation, embryoid bodies at the size of 250–300 µm were embedded into Matrigel and cultured in DMEM/F12 basal media (Thermo Fisher Scientific) supplemented with glutamine (Thermo Fisher Scientific), non-essential amino acids (Thermo Fisher Scientific), 100× ITS-G (Thermo Fisher Scientific), (Basal Media) 3 µM CHIR99021 (Sigma-Aldrich), and 0.5 µM LDN193189 (Sigma-Aldrich). On day 3, human recombinant basic fibroblast growth factor

(bFGF) (PeproTech) at 10 ng/µl final concentration was added to the media. Subsequently, on day 5, the concentration of bFGF was reduced to 5 ng/µl and the media was further supplemented with 10 nM retinoic acid (Sigma-Aldrich). The differentiation media, on day 7, was supplemented only with human recombinant Sonic Hedgehog (hShh) (PeproTech) at 34 ng/µl, human recombinant WNT1A (PeproTech) at 20 ng/µl, and 0.5 µM LDN193189. On day 11, the cytokine composition of the media was changed to 10 ng/µl of bFGF and human recombinant hepatocyte growth factor (HGF) at 10 ng/µl (PeproTech). From day 15 onward, the basal media was supplemented with ITS-X (Thermo Fisher Scientific) and human recombinant HGF at 10 ng/µl. The first three days of the differentiation the media were changed on a daily basis, from 3rd to 30th every second day, while from day 30 onward every 3rd day. The organoid approach was evaluated with six hiPSCs lines with independent genetic backgrounds, with more than five independent derivations per line, especially for the control line (CB CD34+) more than 20 derivation, obtaining always similar results. Per derivation and upon embryoid body Matrigel embedding, cultures exhibited high reproducibility. Upon migration and skeletal muscle formation, organoids are occupying whole Matrigel droplet and through generation of additional bulge reach sizes of 4–5 mM. Concomitantly, myogenic progenitors fall off the organoid and generate a sheath of myogenic organoids and muscle fibers at the surface of the culture plate. For all lines, functional myofibers and PAX7-positive myogenic populations could be evaluated. Myogenic populations from different lines exhibit high similarity (Pearson correlation, rho = 0.94–0,95, *Figure 4—figure supplement 3G*).

### Immunocytochemistry

#### Cryosection immunochemistry

Organoids from different stages were fixed on 4% paraformaldehyde overnight at 4°C under shaking conditions, dehydrated (30% sucrose overnight incubation), and embedded in OCT freezing media. Cryosections were acquired on a Leica CM3050s Cryostat. For the immunostaining process, cryosections were rehydrated with PBS, followed by permeabilization once with 0.1% Tween-20 in PBS (rinsed 3× with PBS), and then with 0.1% Triton-X in PBS (rinsed 3× with PBS). Subsequently, the sections were blocked with 1% BSA/10% NGS in PBS for 1 hr at room temperature (RT). Primary antibody incubations were performed from 1 to 2 d at 4°C, with secondary antibody incubations for 2 hr at RT.

#### EdU staining

At 12 wk post differentiation, control organoids were incubated with 2.5 µM BrdU final concentration overnight. To detect EdU, the sections were processed with Click-iT EdU Alexa Fluor 488 cell proliferation kit (Invitrogen) following the manufacturer's instructions. The samples were incubated with secondary antibodies after the click reaction for detecting EdU. *Primary antibodies*: anti-Brachyury (R&D Systems, 1:250), anti-TBX6 (Abcam, 1:200), anti-PAX3 (DHSB, 1:250), anti-PAX7 (DHSB, 1:250), anti-SOX10 (R&D Systems, 1:125), anti-KI67 (Thermo Fisher Scientific, clone SolA15, 1:100), anti-TITIN (DHSB, 9D-10, 1:300), anti-MyHC (DHSB, MF20, 1:300), anti-MYOD1 (Santa Cruz Biotechnologies, clone 5.8A, 1:200), anti-PRDM16 (Abcam, ab106410, 1:200), anti-TFAP2A (DHSB, 3B5, 1:100), anti-dystrophin (Novocastra/Leica Biosystems, clone DY4/6D3, 1:200), anti-Laminin (Sigma-Aldrich, 1:200), anti-FastMyHC (Sigma-Aldrich, clone MY-32, 1:300), anti-M-Cadherin (Cell Signaling Technology, 1:200) anti-SOX2 (Thermo Fisher Scientific, clone Btjce, 1:100), anti-CD44 (eBioscience, clone IM7, 1:100), and anti-Fibrillin1 (Invitrogen, clone 11C1.3, 1:100).

#### Secondary antibodies

Alexa Fluor 647 AffiniPure Fab Fragment Goat Anti-Mouse IgM, µ Chain specific (Jackson ImmunoResearch Laboratories, 1:100), Rhodamine RedTM-X (RRX) AffiniPure Goat Anti-Mouse IgG, Fcγ Subclass 1 specific (Jackson ImmunoResearch Laboratories, 1:100), Alexa Fluor 488 AffiniPure Goat Anti-Mouse IgG, Fcγ subclass 2a specific (Jackson ImmunoResearch Laboratories, 1:100), Alexa Fluor 488, Goat Anti-Rat IgG (H+L) Cross-Adsorbed Secondary Antibody (Thermo Fisher Scientific, 1:500), Alexa Fluor 488, Donkey Anti-Mouse IgG (H+L) Cross-Adsorbed Secondary Antibody (Thermo Fisher Scientific, 1:500), Alexa Fluor 647, Donkey Anti-Goat IgG (H+L) Cross-Adsorbed Secondary Antibody (Thermo Fisher Scientific, 1:500), Alexa Fluor 488, Donkey Anti-Goat IgG (H+L) Cross-Adsorbed Secondary Antibody (Thermo Fisher Scientific, 1:500), and Alexa Fluor 568, Donkey Anti-Rabbit IgG

(H+L) Cross-Adsorbed Secondary Antibody (Thermo Fisher Scientific, 1:500). Images were acquired on a ZEISS LSM780 inverted confocal microscope.

## Oil O Red staining

For histological visualization of adipocytes within the organoids, Oil O Red Stain kit (Abcam, ab150678) was applied on frozen sections derived from PFA-fixated organoids following the manufacturer's recommended protocol. Organoid sections upon staining with Oil O Red were visualized with an Olympus BX61 upright microscope.

## Flow cytometry

### FACS intracellular staining

Organoids during the 2nd, 5th, 9th, and 14th week of differentiation were dissociated into single cells by incubating them till dissociating at 37°C within papain solution under shaking conditions. Then, the cells were pelleted at 400 × $g$ for 5 min, followed by incubation with TryplE Select for 10 min to ensure dissociation into single cells. Further, the cells were passed through 70 µM (2nd week) to 100 µM (5th, 9th, 14th week) cell strainers to avoid aggregates. For both digesting steps, 10% FBS/DMEM-F12 as digesting deactivation solution was applied to the cells. Then, for intracellular flow cytometric staining analysis the Transcription Factor Buffer set (BD Pharmigen) was applied and the cells were analyzed using flow cytometer (BD Biosciences FACS ARIAII). Primary antibodies used in this study: anti-PAX7, anti- MYOD1, anti-Pax3 in total amount of 400 µg per staining; secondary antibodies: Rhodamine RedTM-X (RRX) AffiniPure Goat Anti-Mouse IgG, Fcγ Subclass 1 specific (Jackson ImmunoResearch Laboratories), Alexa Fluor 488 AffiniPure Goat Anti-Mouse IgG, Fcγ subclass 2a specific (Jackson ImmunoResearch Laboratories) in 1:50 dilution. As isotype controls, Mouse IgG1 kappa Isotype Control (Invitrogen, clone P3.6.2.8.1) and Mouse IgG2a kappa Isotype Control (Invitrogen, clone eBM2a) were used at 400 µg total amount per staining.

### FACS EdU assay

At 15 wk post differentiation, organoids were incubated overnight with 5 µM EdU final concentration. The next day, organoids were dissociated into single cells by incubation at 37°C within papain solution for 1–2 hr, followed by incubation with TryplE Select for 10 min to ensure single-cell dissociation. Then, the dissociated cells were passed through a 70 µm cell culture strainer to remove any remaining aggregates. To detect EdU, the cells were processed with Click- iT EdU Alexa Fluor 488 Flow Cytometry Assay Kit (Invitrogen) according to the manufacturer's instructions and then analyzed using the 488 channel of a BD Biosciences FACSAria Fusion flow cytometer.

### FACS isolation of ITGβ1⁺/CXCR4⁺ myogenic cell population for RNA sequencing

Organoids from Duchenne and Control iPSC lines and during 15th–16th week post differentiation were dissociated into single cells during incubation, with Papain solution dissociation upon gentle shaking (1–2 hr) was observed. To acquire singlets, the cells were filtered through 40 µm cell strainers and upon washing with 1% BSA solution prepared for surface antigen staining. For surface antigen staining, 20 min incubation with the Alexa Fluor 488 anti-human CD29 (BioLegend, clone TS2/16) and PE anti-human CD184[CXCR4] (BioLegend, clone 12G5) was applied together with the corresponding isotype controls: PE Mouse IgG2a, κ Isotype Ctrl antibody (BioLegend, clone MOPC-173) and 488 Mouse IgG1, κ Isotype Ctrl antibody (Invitrogen, clone P3.6.2.8.1). For removing residual antibodies, the cells were washed twice with 1% BSA staining solution and processed by BD Biosciences FACSAria Fusion flow cytometer. Briefly before FACS sorting to discriminate between dead and alive cells, DAPI was added to the samples and then DAPI⁻/CD29⁺ /CXCR4⁺ cell populations were collected into tubes containing RLT buffer supplemented with b-mercaptoethanol to avoid RNA degradation. The FACS gating strategy is further depicted in *Figure 4—figure supplement 3A*.

## Bulk RNA sequencing

### RNA extraction

Total RNA was extracted from single organoids or cultured cells by using the RNAeasy Micro Kit (QIAGEN) according to the manufacturer's instructions. Subsequently, before library preparation, the RNA integrity was evaluated on an Agilent 2100 Bioanalyzer by using the RNA 6000 Pico kit (Agilent). *cDNA library preparation*: For 4 and 8 wk organoids, cDNA library was prepared by using the whole transcriptome Illumina TruSeq Stranded Total RNA Library Prep Kit Gold (Illumina), followed by evaluation on an Agilent 2100 Bioanalyzer by using the DNA 1000 kit. The resulting mRNA library was sequenced as 2 × 75 bp paired-end reads on a NextSeq 500 sequencer (Illumina). For 16 wk and ITGβ1+/CXCR4+ sorted cells, cDNA library was prepared using the whole transcriptome Ovation Solo RNA seq Library Preparation Kit (TECAN, NuGEN), followed by evaluation on an Agilent 2100 Bioanalyzer by using the DNA 1000 Chip. The resulting mRNA library was sequenced as 1 × 150 bp single reads on a HiSeq 3000 sequencer (Illumina).

## Bulk RNA-seq bioinformatic analysis

Sequenced reads were aligned to the human reference genome (hg38) with TopHat2 (version 2.1.1), and the aligned reads were used to quantify mRNA expression by using HTSeq-count (version 0.11.2). DESeq2 (*Love et al., 2014*) was used to identify differentially expressed genes (DEGs) across the samples. ITGβ1+/CXCR4+ organoid-derived myogenic cell populations were compared to already available transcriptomic dataset of human fetal muscle progenitors (GSM2328841-2) (*Hicks et al., 2018*).

## scRNA sequencing

### Sample and cDNA library preparation

Single cells were acquired upon incubation for 1 hr with solution containing papain and EDTA. Upon dissociation, the cell number and viability were estimated. Then, cells were resuspended in a solution containing 0.5% BSA in PBS to reach a concentration of 390 cells per μl. The cDNA library was prepared using the Chromium Single Cell 3′ Reagent Kits (v3): Single Cell 3′ Library & Gel Bead Kit v3 (PN-1000075), Single Cell B Chip Kit (PN-1000073), and i7 Multiplex Kit (PN-120262) (10x Genomics) according to the manufacturer's instructions. Then, the cDNA library was run on an Illumina HiSeq 3000 as 150 bp paired-end reads.

## Single-cell RNA-seq bioinformatic analysis

Sequencing data were processed with UMI-tools (version 1.0.0), aligned to the human reference genome (hg38) with STAR (version 2.7.1a), and quantified with Subread featureCounts (version 1.6.4). Data normalization and further analysis were performed using Seurat (version 3.1.3, *Stuart et al., 2019*). For initial quality control of the extracted gene–cell matrices, cells were filtered with parameters low threshold = 500, high threshold = 6000 for number of genes per cell (nFeature_RNA), high threshold = 5 for percentage of mitochondrial genes (percent.mito), and genes with parameter min. cells = 3. Filtered matrices were normalized by the LogNormalize method with scale factor = 10,000. Variable genes were found with parameters of selection.method = 'vst,' nfeatures = 2000, trimmed for the genes related to cell cycle (KEGG cell cycle, hsa04110) and then used for principal component analysis (PCA). Statistically significant principal components were determined by the JackStraw method and the first five principal components were used for nonlinear dimensional reduction (tSNE and UMAP) and clustering analysis with resolution = 0.2. Monocle3 (version 0.2.0, *Cao et al., 2019*) was used for pseudotime trajectory analysis. The data matrix of Seurat objects (assays[["RNA"]]@counts) was imported to the Monocle R package, then dimensionality reduction with the PCA method with parameters max_components = 2 was performed and then cluster_cells, learn_graph and order_cells functions were performed subsequently. Organoid-derived myogenic progenitors were compared to already available transcriptomic dataset of adult satellite cells (GSE130646) (*Rubenstein et al., 2020*).

For integrative analysis, we used the Seurat (version 4, *Hao et al., 2021*) package. For each dataset, cells with >6000 or <300 detected genes, as well as those with mitochondrial transcripts proportion >5–10%, were excluded. For finding anchors between control and DMD datasets using 5000 anchors and regressing out cell cycle genes, sequencing depth and stress-related genes were carried out before integration.

Developmental score was calculated as described in *Xi et al., 2020*. Briefly, we used the "AddModuleScore" function to calculate the embryonic and adult score using a list of DEGs between adult and embryonic myogenic progenitor clusters. DEGs were selected from the Supplementary information in *Xi et al., 2020* table mmc3. The developmental score was further calculated by subtracting embryonic from the adult score. Embryonic and fetal datasets were filtered using a low threshold = 500 and adult datasets were filtered using a low threshold = 250 for genes per cell. In addition, scaling the data ("S.Score,' 'G2M.Score,' 'Stress,' and 'total Count' to the vars.to.regress' argument) was used to regress out the effects of the cell cycle, dissociation-related stress, as well as cell size/sequencing depth to all datasets. Myogenic subpopulation of SMO and adult satellite cell datasets were selected by 'PAX7' expression. Embryonic and fetal (weeks 5–18) and adult satellite cell (years 7, 11, 34, and 42) scRNAseq data are from *Xi et al., 2020* (GSE147457), adult satellite cell (year 25) scRNAseq data are from *Rubenstein et al., 2020* (GSE130646).

## Cell–cell communication analysis

To investigate cell–cell communications among activated, mitotic and dormant myogenic progenitors, fibroadipogenic and neural progenitors, and myofiber-related clusters from 12 wk organoids, the CellChat R package (*Jin et al., 2021*) was applied.

## qPCR expression analysis

By pooling three organoids per sample, total RNA was extracted using the RNAeasy Mini Plus Kit (QIAGEN). For first-strand cDNA synthesis, the High Capacity RNA to cDNA Kit (Applied Biosystems) was applied using as template 2 µg of total RNA. For setting qPCR reactions, the GoTaq qPCR Master Mix (Promega) was used with input template 4 ng cDNA per reaction while the reaction was detected on a CFX 96 Real-Time PCR detection system (Bio-Rad). The relative quantification (ΔCT) method was applied for detecting changes in gene expression of pluripotent, neural tube, neural crest, posterior/anterior somitic, and dermomyotomal markers, between different time points along the differentiation. qPCR primers applied for each marker for evaluating organoid development are listed in *Supplementary file 1a*.

## Diffusion map analysis

By pooling 3–5 organoids per sample, total RNA was extracted using the RNAeasy Mini Plus Kit (QIAGEN) and further processed as described under qPCR expression analysis. Normalized Ct values of the selected genes for each sample were used as input for generating eigenvector values using the destiny package (*Angerer et al., 2016*). Then, all samples were ordered by their Diffusion Component 1 at specific timepoints during early culture development (day 2–day 11). Ct values >30 were not considered for subsequent analysis. qPCR primers applied for each marker for diffusion map analysis are listed in *Supplementary file 1b*.

## Transmission electron microscopy (TEM)

Skeletal muscle organoids were fixed for 4 hr at RT in 2.5% glutaraldehyde (Sigma-Aldrich) in 0.1 M cacodylate buffer pH 7,4 (Sciences Services, Germany), subsequently washed in 0.1 M cacodylate buffer pH 7,4, post-fixed for 2 hr at RT in 1% osmium tetroxide (Sciences Services, Germany) in 0.1 M cacodylate buffer pH 7,4, dehydrated stepwise in a graded ethanol series, and embedded in Epon 812 (Fluka, Buchs, Switzerland). Ultrathin sections (70 nm, ultramicrotome EM UC7, Leica, Wetzlar, Germany) were afterward stained for 30 min in 1% aqueous uranyl acetate (Leica, Germany) and 20 min in 3% lead citrate (Leica). TEM images were acquired with a 200 kV TEM JEM 2100Plus (Jeol, Japan), transmission electron microscope.

## Second harmonic generation (SHG) imaging using multi-photon microscopy

A TriM Scope II multi-photon system from LaVision BioTec was used to visualize skeletal muscle fiber organization inside organoids and distinct sarcomeres. The microscope setup is a single-beam instrument with an upright Olympus BX51 WI microscope stand equipped with highly sensitive non-descanned detectors close to the objective lens. The TriM Scope II is fitted with a Coherent Scientific Chameleon Ultra II Ti:Sapphire laser (tuning range 680–1080 nm) and a Coherent Chameleon

Compact OPO (automated wavelength extension from 1000 nm to 1600 nm). A 20× IR objective lens (Olympus XLUMPlanFl 20×/1.0W) with a working distance of 2.0 mm was used. Muscle fiber SHG signals were detected in forward direction using TiSa light at 850 nm, a 420/40 band pass filter, and a blue-sensitive photomultiplier (Hamamatsu H67080-01). 3D images were acquired and processed with LaVision BioTec ImSpector Software.

## Electrophysiology

### Current measurement

Membrane currents were measured at ambient temperature (22–24°C) using standard whole-cell patch clamp software ISO2 (MFK, Niedernhausen, Germany). Cells were voltage-clamped at a holding potential of –90 mV, that is, negative to EnAChR, resulting in inward $Na^+$ currents. Every 10 s, voltage ramps (duration 500 ms) from –120 mV to +60 mV were applied to assess the stability of the recording conditions and generate I/V curves (membrane currents in response to depolarizing voltage ramps are shown as downward deflections). Signals were filtered (corner frequency, 1 kHz), digitally sampled at 1 kHz, and stored on a computer equipped with the hardware/software package ISO2 for voltage control, data acquisition, and data analysis. Rapid exposure to a solution containing acetylcholine was performed by means of a custom-made solenoid-operated flow system permitting a change of solution around an individual cell with a half-time of about 100 ms. For measurements, cells devoid of contact with neighboring cells were selected. Cells originated from organoids at week 8.

### Fluorescence microscopy and imaging

To monitor changes in $[Ca^{2+}]i$, skeletal muscle cells were transiently transfected with pcDNA3[Twitch-2B] (Addgene, 49531) (0.25 μg per 35 mm culture dish). Skeletal muscle cells were transfected using either poly-ethyleneimine (PEI) or Lipofectamine (Invitrogen) according to the manufacturer's instructions. Prior to experiments, cells were seeded on sterile, poly-L-lysine-coated glass coverslips and analyzed for 48 hr after transfections. All experiments were performed using single cells at ambient temperature. Fluorescence was recorded using an inverted microscope (Zeiss Axiovert 200, Carl Zeiss AG, Göttingen, Germany) equipped with a Zeiss oil immersion objective (100×/1.4), a Polychrome V illumination source, and a photodiode-based dual-emission photometry system suitable for CFP/YFP-FRET (FEI Munich GmbH, Germany). For FRET measurements, single cells were excited at 435 nm wavelength with light pulses of variable duration (20–50 ms; frequency: 5 Hz) to minimize photo-bleaching. Corresponding emitted fluorescence from CFP (F480 or FCFP) or from YFP (F535 or FYFP) was acquired simultaneously, and FRET was defined as the ratio of FYFP/FCFP. Fluorescent signals were recorded and digitized using a commercial hardware/software package (EPC10 amplifier with an integrated D/A board and Patch-master software, HEKA, HEKA Elektronik, Germany). The individual FRET traces were normalized to the initial ratio value before agonist application (FRET/FRET0).

### Solutions and chemicals

For FRET measurements, an extracellular solution of the following composition was used (mmol/l): NaCl 137; KCl 5.4; $CaCl_2$ 2; $MgCl_2$ 1.0; HEPES/NaOH 10.0, pH 7.4. For whole-cell measurements of membrane currents, an extracellular solution of the following composition was used (in mmol/l): NaCl 120, KCl 20, $CaCl_2$ 0.5, $MgCl_2$ 1.0, HEPES/NaOH 10.0 (pH 7.4). The pipette solution contained (in mmol/l): K-aspartate 100, KCl 40, NaCl 5.0, $MgCl_2$ 2.0, Na2ATP 5.0, BAPTA 5.0, GTP 0.025, and HEPES/KOH 20.0, pH 7.4. Standard chemicals were from Merck. EGTA, HEPES, Na2ATP, GTP, and acetylcholine chloride were from Sigma-Aldrich.

## Transplantation experiments

### Flow cytometry

The organoid culture was dissociated using TrypLe (Thermo Fisher Scientific, 12563011) and filtered through 70–40 μm cell strainer. Single-cell suspension was stained with PE anti-human CD82 Antibody (BioLegend, 342103) in 2% BSA and 2 mM EDTA and sorted using FACS sorter (Beckman Coulter, MoFlo Astrios Zellsortierer). For gating, unstained single cells from the same organoid culture were used as a baseline control. Sorted cells were further processed for transplantation experiments.

## Transplantation of skeletal muscle progenitors into murine TA muscle

Twelve-week-old organoids were dissociated and single cells were sorted using FACS (CD82+). Cell transplantation was carried out as described before (*Alexander et al., 2016*; *Marg et al., 2019*; *Al Tanoury et al., 2020*). Briefly, 24 hr before transplantation, Cardiotoxin (10 µl CTX, 40 ng/ml, Sigma-Aldrich, 217503-1MG) was injected into TA of 2–3-month-old male HsdCpb:NMRI-*Foxn1*$^{nu}$ mice. Under anesthesia, $1 * 10^5$ cells were injected into the TA muscle on one side. Six weeks after transplantation, the mice were killed and the TA was fixed and sliced using cryosections. Immunofluorescence analyses were carried out using recombinant Anti-Lamin A + Lamin C antibody and anti-dystrophin antibody.

All animal experiments were approved by the local authorities (81-02.04.2020.A476, Ruhr University Bochum) and performed in accordance with the guidelines for Ethical Conduct in the Care and Use of Animals.

## Cryosection immunochemistry

TA from mice was fixed on 4% paraformaldehyde overnight at 4°C and kept in increasing sucrose solutions (10, 20, and 30%) at 4°C until the tissue sinked down (usually 24 hr) and embedded in Tissue-Tek O.C.T. Compound media. Cross-sections of TA with 10–20 µm thickness were acquired on CryoStar NX50 (Thermo Scientific). Sections were stored on objectives at –20°C.

Cryosections were rehydrated with PBS, followed by an antigen retrieval, an AffiniPure Fab Fragment Goat Anti-Mouse IgG treatment to prevent unspecific bindings during the staining process and permeabilized once with 1% (vol/vol) Triton-X100 and 125 mM glycine in PBS (rinsed 3× with PBS). Subsequently, the sections were blocked with 5% BSA/10% NGS in PBS for 1 hr at room temperature. Primary antibody incubations were performed for 1 d at 4°C, and secondary antibody incubations for 2 hr at room temperature.

Primary antibodies: anti-dystrophin (Leica, NCLDYS1, 1:20), recombinant Anti-Lamin A + Lamin C antibody (Abcam, ab108595, 1:150). Secondary antibodies: Goat Anti-Rabbit IgG (H+L) Highly Cross-Adsorbed Secondary Antibody, Alexa Fluor Plus 488 (1:1000), Goat Anti-Mouse IgG (H+L) Cross-Adsorbed Secondary Antibody, Alexa Fluor 568 (1:1000).

Images were acquired on a Zeiss Scan.Z1. Images were processed using Zen Lite Blue version 4.0.3.

## Statistics

All statistical analyses were conducted using GraphPad Prism6 software. For qPCR analysis, one-way ANOVA with Tukey's multiple-comparisons test for each marker was performed. For the FACS intracellular staining quantification, one-way ANOVA with Sidak's multiple-comparisons test between the different time points was performed. Significance asterisks represent *$p<0.05$, **$p<0.01$, ***$p<0.001$, ****$p<0.0001$, ns: not significant.

## Acknowledgements

We are grateful to Drs. Karl Köhrer, Tobias Lautwein, Patrick Petzsch, and Thorsten Wachtmeister, Genomics & Transcriptomics Laboratory, Heinrich-Heine-University Düsseldorf for performing single-cell and bulk RNAseq experiments with their Illumina HiSeq platform and data provision. We are further grateful to Dr. Oliver Griesbeck for the pcDNA3[Twitch-2B] plasmid. We would like to thank Ingrid Gelker and Martina Sinn, Max Planck Institute Münster as well as Eva-Maria Konieczny, Rana Houmany, and Boris Burr, Ruhr-University Bochum, for their technical assistance. We thank Dr. Johnny Kim, Max Planck Institute Bad Nauheim, for discussions and Dr. Elisabeth Stevens, *English Scientific*, Düsseldorf, for scientific editing of the manuscript. Electron microscopy experiments were supported by the Deutsche Forschungsgemeinschaft SFB 944. We thank Drs. George Q Daley and Thorsten Schlaeger, Boston Children's Hospital, for providing the Duchenne Muscular Dystrophy patient-derived iPS cell lines DMD-iPS1 and BMD-iPS1 in the course of our study. Our study was supported by research grants from FoRUM F873-16, Medical Faculty, Ruhr University Bochum, from Deutsche Gesellschaft für Muskelkranke e.V. (DGM Foundation), Freiburg, Georg E und Marianne Kosing-Stiftung, Deutsches Stiftungszentrum, Essen and Deutsche Duchenne Stiftung, Duchenne Deutschland e.V.

# Additional information

## Competing interests

Ji Hun Yang: is partially employed by Next & Bio Inc. The other authors declare that no competing interests exist.

## Funding

| Funder | Grant reference number | Author |
|--------|------------------------|--------|
| Deutsche Duchenne Stiftung, Duchenne Deutschland e.V., | DFP2018 | Beate Brand-Saberi Holm Zaehres |
| Deutsche Gesellschaft für Muskelkranke | Za1/1 | Holm Zaehres |

The funders had no role in study design, data collection and interpretation, or the decision to submit the work for publication. Open access funding provided by Max Planck Society.

## Author contributions

Lampros Mavrommatis, Conceptualization, Resources, Data curation, Formal analysis, Validation, Investigation, Visualization, Methodology, Writing – original draft, Project administration; Hyun-Woo Jeong, Data curation, Software, Formal analysis, Investigation, Methodology, Writing – original draft; Urs Kindler, Data curation, Software, Formal analysis, Investigation, Methodology; Gemma Gomez-Giro, Marie-Cecile Kienitz, Martin Stehling, Olympia E Psathaki, Dagmar Zeuschner, M Gabriele Bixel, Dong Han, Gabriela Morosan-Puopolo, Data curation, Formal analysis, Investigation, Methodology; Daniela Gerovska, Ji Hun Yang, Marcos J Arauzo-Bravo, Data curation, Software, Formal analysis; Jeong Beom Kim, Resources, Data curation, Software, Formal analysis; Jens C Schwamborn, Resources, Data curation, Formal analysis; Stephan A Hahn, Resources, Data curation, Formal analysis, Investigation; Ralf H Adams, Resources, Supervision; Hans R Schöler, Resources, Data curation, Supervision; Matthias Vorgerd, Resources, Data curation, Formal analysis, Supervision; Beate Brand-Saberi, Resources, Supervision, Writing - review and editing; Holm Zaehres, Conceptualization, Resources, Data curation, Formal analysis, Supervision, Funding acquisition, Validation, Investigation, Visualization, Methodology, Writing – original draft, Project administration, Writing - review and editing

## Author ORCIDs

Hyun-Woo Jeong http://orcid.org/0000-0002-6976-6739
Urs Kindler http://orcid.org/0000-0001-9676-6323
Dagmar Zeuschner http://orcid.org/0000-0002-6712-0192
Jeong Beom Kim http://orcid.org/0000-0001-6230-8826
Marcos J Arauzo-Bravo http://orcid.org/0000-0002-3264-464X
Jens C Schwamborn https://orcid.org/0000-0003-4496-0559
Stephan A Hahn https://orcid.org/0000-0003-0855-9741
Ralf H Adams http://orcid.org/0000-0003-3031-7677
Holm Zaehres http://orcid.org/0000-0001-8062-8428

## Ethics

All animal experiments were approved by the local authorities (81-02.04.2020.A476, Ruhr University Bochum) and performed in accordance with the guidelines for Ethical Conduct in the Care and Use of Animals.

Reviewer #1 (Public Review): https://doi.org/10.7554/eLife.87081.3.sa1
Reviewer #2 (Public Review): https://doi.org/10.7554/eLife.87081.3.sa2
Author Response https://doi.org/10.7554/eLife.87081.3.sa3

# Additional files

## Supplementary files
• Supplementary file 1. Tables of qPCR primer pairs. (**a**) qPCR primer pairs applied to detect relative expression of key markers during skeletal muscle organoid development. (**b**) qPCR primer pairs applied for diffusion map analysis of early skeletal muscle organoid development.

• MDAR checklist

## Data availability
RNA sequencing datasets produced in this study are deposited in the Gene Expression Omnibus (GEO) under accession code GSE147514. Data and code availability section' or you can replace it with "All the analysis scripts are available from vignettes of original software webpage of Seurat (https://satijalab.org/seurat/) and Monocle (https://cole-trapnell-lab.github.io/monocle3/). No custom code or mathematical algorithm other than variable assignment was used in this study. To review GEO accession GSE147514:Go to https://www.ncbi.nlm.nih.gov/geo/query/acc.cgi?acc=GSE147514.

The following dataset was generated:

| Author(s) | Year | Dataset title | Dataset URL | Database and Identifier |
|---|---|---|---|---|
| Jeong H-W, Mavrommatis L, Zaehres H | 2023 | Transcriptome profiling of human skeletal muscle organoids | https://www.ncbi.nlm.nih.gov/geo/query/acc.cgi?acc=GSE147514 | NCBI Gene Expression Omnibus, GSE147514 |

The following previously published datasets were used:

| Author(s) | Year | Dataset title | Dataset URL | Database and Identifier |
|---|---|---|---|---|
| Xi et al. | 2020 | Single Cell RNA-Sequencing of Human Limb Skeletal Muscle across Development and Myogenic Culture from Pluripotent Stem Cells | https://www.ncbi.nlm.nih.gov/geo/query/acc.cgi?acc=GSE147457 | NCBI Gene Expression Omnibus, GSE147457 |
| Rubinstein et al. | 2020 | Single-cell transcriptional profiles in human skeletal muscle | https://www.ncbi.nlm.nih.gov/geo/query/acc.cgi?acc=GSE130646 | NCBI Gene Expression Omnibus, GSE130646 |
| Eskin A | 2016 | 1_Fresh_W17_0_Imm_Sort_S6 | https://www.ncbi.nlm.nih.gov/geo/query/acc.cgi?acc=GSM2328841 | NCBI Gene Expression Omnibus, GSM2328841 |

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
