## [Editor Report · eLife assessment]

The authors develop a cell culture system for studies of muscle tissue development and homeostasis. They **convincingly** validate a novel 3D cell model. Their thorough molecular and functional characterization will make this **useful** for future workers in the field.

---

## [Referee Report · Reviewer #1 (Public Review)]

The authors aimed to establish a cell culture system to investigate muscle tissue development and homeostasis. They successfully developed a complex 3D cell model and conducted a comprehensive molecular and functional characterization. This approach represents a critical initial step towards using human cells, rather than animals, to study muscular disorders in vitro. Although the current protocol is time-consuming and the fetal cell model may not be mature enough to study adult-onset diseases, it nonetheless provides a valuable foundation for future disease modeling studies using isogenic iPSC lines or patient-derived cells with specific mutations. The manuscript does not explore whether or how this stem cell model can advance our understanding of muscular diseases, which would be an exciting avenue for future research. Overall, the detailed protocol presented in this paper will be useful for informing future studies and provide a valuable resource to the stem cells community. Future work could focus on disease modeling using isogenic iPSC lines or patient-derived cells.

---

## [Referee Report · Reviewer #2 (Public Review)]

This paper illustrates that PSCs can model myogenesis in vitro by mimicking the in vivo development of the somite and dermomyotome. The advantages of this 3D system include (1) better structural distinctions, (2) the persistence of progenitors, and (3) the spatial distribution (e.g. migration, confinement) of progenitors. The finding is important with the implication in disease modeling. Indeed the authors tried DMD model although it suffered the lack of deeper characterization.

The differentiation protocol is based on a current understanding of myogenesis and is compelling. They characterized the organoids in depth (e.g. many time points and immunofluorescence). The evidence is solid.

---

## [Author Response]

The following is the authors’ response to the original reviews.

We want to thank you for organizing the review process a of our manuscript ‘Human skeletal muscle organoids model fetal myogenesis and sustain uncommitted PAX7+ myogenic progenitor’ for eLife and the reviewers for providing their criticisms.

We have changed some Figures within the manuscript and added two new Supplementary Figures as outlined below

**Reviewer #1 (Public Review):**
The authors aimed to establish a cell culture system to investigate muscle tissue development and homeostasis. They successfully developed a complex 3D cell model and conducted a comprehensive molecular and functional characterization. This approach represents a critical initial step towards using human cells, rather than animals, to study muscular disorders in vitro. Although the current protocol is time-consuming and the fetal cell model may not be mature enough to study adult-onset diseases, it nonetheless provides a valuable foundation for future disease modelling studies using isogenic iPSC lines or patient-derived cells with specific mutations. The manuscript does not explore whether or how this stem cell model can advance our understanding of muscular diseases, which would be an exciting avenue for future research. Overall, the detailed protocol presented in this paper will be useful for informing future studies and provides an important resource to the stem cells community. The inclusion of data on disease modelling using isogenic iPSC lines or patient-derived cells would further enhance the manuscript's impact.

We agree, that data on disease modelling using patient-derived cells would further enhance the manuscript's impact. The manuscript in its current form should present our skeletal muscle organoid differentiation protocol to the community with a focus of the developmental processes which are mimiced by this model. We are not aiming to disease model e.g. LGMD or Duchenne within the context of this study. Our protocol is just the starting point of us and others to use this organoid protocol for skeletal muscle disease modelling in further studies. We already have a study of Duchenne musculular dystrophy modelling using our organoid system under way.

**Reviewer #2 (Public Review):**
This paper illustrates that PSCs can model myogenesis in vitro by mimicking the in vivo development of the somite and dermomyotome. The advantages of this 3D system include (1) better structural distinctions, (2) the persistence of progenitors, and (3) the spatial distribution (e.g. migration, confinement) of progenitors. The finding is important with the implication in disease modeling. Indeed the authors tried DMD model although it suffered the lack of deeper characterization.The differentiation protocol is based on a current understanding of myogenesis and compelling. They characterized the organoids in depth (e.g. many time points and immunofluorescence). The evidence is solid, and can be improved more by rigorous analyses and descriptions as described below.Major comments:1. Consistency between different cell lines.I see the authors used a few different PSC lines. Since organoid efficiency differ between lines, it is important to note the consistency between lines.1. Heterogeneity among each organoidLet's say authors get 10 organoids in one well. Are they similar to each other? Does each organoid possess similar composition of cells? To determine the heterogeneity, the authors could try either FACS or multiple sectioning of each organoid.

Concerning the raised issue of consistency between different PSC lines we stated under Material and Methods that skeletal muscle organoids were generated from six hiPSC lines: CB-CD34 iPSC, DMD iPSC, DMD_iPS1, BMD_iPS1, LGMD2A iPSC, LGMD2A-isogenic iPSC. We have evaluated the organoid approach with six hiPSC lines with independent genetic backgrounds with more than 5 independent derivations per line, for the control line (CB CD34+) with more than 20 derivations. At the time of creating the first preprint in 2020 our reported protocol was based on about 45 independent differentiation inductions.

The heterogeneity among each organoid is a valid point, however very cumbersome to address with FACS or multiple sectioning.

We have now addressed the heterogeneity of organoids within a line and the consistency of organoids between different lines by diffusion map analysis for early organoid stages and further single cell RNA seq analyses for mature stages and include this data as Figure 4 – figure supplement 6.

1. Consistency of Ach current between organoids.Related to comment 2, are the currents consistent between each organoid? How many organoids were recorded in the figures? Also, please comment if the current differ between young and aged organoids.

The acetylcholine (ACh)-induced changes in holding currents in Figure 3K are representative recordings with n = 6. The further recordings in Figure 3 – Figure Supplemental 3 for organoids derived from three additional lines, were also recorded with n = 6. Cells were taken for electrophysiological characterization in all analyses from 8 weeks organoids.

1. Communication between neural cells and muscle?The authors did scRNAseq, but have not gone deep analysis. I would recommend doing Receptorligand mapping and address if neural cells and muscle are interacting.

We are now providing a characterization of the cell-cell communication network for all clusters at week 12 of human skeletal muscle organoid development as the new Figure 4 – figure supplement 5.

1. More characterization of DMD organoids.One of the key applications of muscle organoids is disease model. They have generated DMD muscle organoids, but rarely characterized except for currents. I recommend conducting immunofluorescence of DMA organoids to confirm structure change. Very intriguing to see scRNAseq of DMD organoids and align with disease etiology.

We agree, that data on disease modelling using DMD patient-derived cells would further enhance the manuscript's impact. The manuscript in its current form should present our skeletal muscle organoid differentiation protocol to the community with a focus of the developmental processes which are mimiced by this model. We already have a study of Duchenne muscular dystrophy modelling using our organoid system under way.

1. More characterization of engraft.Authors could measure the size of myotube between mice and human.

We have quantitatively evaluated the myotubes in the transplantation experiment illustrated in Figure 4I,J. The mean diameter is 41+/-6 µm for the human and 63+/-7 µm for the mice fibers (n = 15 each). See Author response image 1.

Does PAX7+ satellite cell exist in engraft? To exclude cell fusion events make up the observation, I recommend to engraft in GFP+ immunodeficient mice. Could the authors comment how long engraft survive.

We would claim satellite cells within our engrafts with the DAPI-blue nuclei surrounded by green human lamin A/C as in Author response image 2. We have analysed all our mice six weeks post transplantation for engrafting similar to other groups in the field.

**Author response image 2. sa3fig2:** 

**Reviewer #1 (Recommendations For The Authors):**
The manuscript ends abruptly with the mouse transplantation experiment that appears a bit preliminary. It basically shows that cells survive but functional (or ultrastructural) integration is not shown. Suggest clarifying motivation and interpretation of the in vivo data.

Back in 2020 our manuscript had already passed detailed review processes whereby we struggled by not providing any in vivo data concerning repopulation of our progenitor cells. Coming from the human pluripotent stem cell biology field we have never completely understood the value of this hybrid experiments to test human cells in mouse again.

For the current version, we have then taken additional efforts to transplant our progenitor cells into injured skeletal muscle cells similarly to other groups in the field (Alexander et al., 2016, Marg et al., 2019, Tanoury et al., 2020) (Figure 4I,J). A proof that 3D-derived progenitor cells have a clear repopulation advantage over progenitor cells derived in a 2D protocol would go beyond what can be done within the scope of our study. We are still mainly basing our claims on the extended bulk and single RNA seq comparison to progenitor cells obtained by others. However, to address the demand of several experts to test our cells also in vivo, we can also provide in vivo data in the current manuscript version.

Within the Discussion we are suggesting further evaluations using these transplantations: It would be of interest for future studies to investigate whether increased engraftment can be achieved in 3D protocols (Faustino Martins et al., 2020; Shahriyari et al., 2022; ours) versus 2D patterned progenitor cells.

**Reviewer #2 (Recommendations For The Authors):**
Minor comments:1. Plot CD82 gene on UMAP of Figure 4

We had provided a CD82 scRNAseq analysis within the t-SNE plots of Figure 3 – figure supplement 1, which is demonstrating, that CD82-positive cells almost exclusively overlap with Pax7-positive cells, being a subcluster of them. We agree, that the reader will benefit from this further analysis and we are now providing in Author response image 3 additional CD82 and Pax7 UMAP plots on the myogenic progenitor / satellite cell clustering analysis of Figure 4F within the new Figure 4 – figure supplement 4E.

**Author response image 3. sa3fig3:** 

1. Immunofluorescence of CD82 in organoids

We have tried CD82 immunofluorescence analysis on our organoids but are not very satisfied with the technical outcome. The available CD82 antibody seems to be primarily suited for FACS analysis and not for immunohistochemistry on slices.

1. Change red-green color of the heatmap. Color-blind person cannot see it well

We have changed all heatmaps to yellow-purple in the main Figure 2G and the Supplemental Figures S2.1 and S3.1..